# Blending Supervised and Reinforcement Fine-Tuning with Prefix Sampling

**Zeyu Huang** [1]   **Tianhao Cheng** [2]   **Zihan Qiu** [3]   **Zili Wang** [4]   **Yinghui Xu** [2]   **Edoardo Ponti** [1]   **Ivan Titov** [1 5]

## Abstract

Existing LLMs-post-training techniques are broadly categorized into supervised fine-tuning (SFT) and reinforcement fine-tuning (RFT). Each paradigm presents a distinct trade-off: (1) SFT excels at mimicking demonstration data, but can lead to problematic generalization as a form of behavior cloning. (2) Conversely, RFT can significantly enhance a model's performance but is prone to learning unexpected behaviors, and its performance is sensitive to the initial policy. In this paper, we propose a unified view of these methods and introduce Prefix-RFT, a hybrid approach that synergizes learning from both demonstration and exploration. Using mathematical reasoning problems as a test bed, we empirically demonstrate that Prefix-RFT is simple yet effective. Not only does it surpass the performance of standalone SFT and RFT, but it also outperforms parallel mixed-policy RFT methods. Our analysis highlights the complementary nature of SFT and RFT, validating that Prefix-RFT effectively harmonizes them. Further ablation studies confirm the method's robustness to variations in the quality and quantity of demonstration data.

## 1. Introduction

LLM post-training is primarily accomplished through two distinct paradigms: supervised fine-tuning (SFT) and reinforcement fine-tuning (RFT). SFT adapts pre-trained models by continuing to train them on curated datasets of labeled examples. Its strength lies in its simplicity: mimicking the "correct" demonstrations. It is thus highly effective for adapting models to various downstream tasks (Wei et al., 2022; Zhu et al., 2025; Huang et al., 2024; Peng et al., 2023; Huang et al., 2025; Yang et al., 2026). However, SFT is fundamentally a form of behavioral cloning and thus probably

[1]ILCC, University of Edinburgh [2]Fudan University [3]Qwen Team, Alibaba Group [4]StepFun [5]ILLC, University of Amsterdam. Correspondence to: Zeyu Huang <zeyu.huang@ed.ac.uk>.

*Proceedings of the 43 rd International Conference on Machine Learning*, Seoul, South Korea. PMLR 306, 2026. Copyright 2026 by the author(s).

leads to a problematic generalization and robustness (Chu et al., 2025a; Chen et al., 2025a; Xie et al., 2024).

Reinforcement fine-tuning (RFT) has been pivotal in overcoming the limitations of SFT and further elevating model capabilities (Hu et al., 2025; Xie et al., 2025). Recent large reasoning models, such as OpenAI-o1 (Jaech et al., 2024) and DeepSeek-R1 (Guo et al., 2025), demonstrated the promise of this approach and have effectively solved problems previously considered intractable, such as competition-level math (Li et al., 2024) and coding problem (Jain et al., 2024). Despite these successes, the RFT paradigm is not uncontentious and faces its own challenges: **(1)** The learning signal from rewards is often sparse; for complex, multi-step tasks, it is difficult to assign credit to the specific tokens that lead to a successful outcome, resulting in unexpected behaviors like language mixing after training (Guo et al., 2025; Yuan et al., 2025). **(2)** Moreover, its effectiveness is highly dependent on the strength of the initial policy (Yue et al., 2025a; Zhao et al., 2025). The process arguably refines and aligns existing capabilities rather than instilling new knowledge (Liu et al., 2025d), leading some work to question whether RL can truly raise a model's intrinsic capability ceiling (Chu et al., 2025b; Liu et al., 2025a; Yue et al., 2025b; Cheng et al., 2025). The gains from RFT, while significant, may stem from perfecting what the model has already learned during pre-training and SFT (Wang et al., 2025; Gandhi et al., 2025).

In total, SFT provides crucial dense supervision for injecting knowledge that a model cannot discover on its own. RFT, by contrast, targets actual competence but is tethered to the model's capabilities. This establishes their *core complementarity*: SFT serves as the mechanism for expanding the model's knowledge boundary, elevating RFT's capability ceiling, while RFT provides the goal-oriented training objective necessary to steer the model from behavioral cloning towards robust problem-solving. This motivates our central research question: *How can we develop a framework that formally integrates the process supervision of SFT with the goal-oriented optimization of RFT?*

To bridge this gap, we first present a unified view of SFT and RFT, suggesting that they share a consistent optimization structure. We then introduce *Prefix Reinforcement Finetuning* (**Prefix-RFT**) as a hybrid post-training approach that

incorporates offline demonstration datasets into RFT training. Specifically, we sample a prefix from the demonstration and task the policy with generating its continuation. This composite sequence—an off-policy prefix followed by the on-policy continuation—is then treated as a trajectory and used alongside standard model rollouts in the RFT update step. The core intuitions behind Prefix-RFT are twofold: (1) Compared to RFT, a high-quality prefix serves as a powerful guiding mechanism for exploration. If a hybrid trajectory yields a higher reward, the corresponding prefix is naturally reinforced into the model. (2) Compared to SFT, Prefix-RFT keeps RFT's problem-solving training objective. Meanwhile, by providing only the initial part of the solution, Prefix-RFT grants the model constrained autonomy: it starts by following a promising path but still has the flexibility to discover a superior continuation, thus leveraging demonstration data for guidance without being rigidly constrained.

We choose math reasoning problems as the test bed for our proposed method. Despite its simplicity, our empirical results demonstrate that the Prefix-RFT outperforms naïve SFT, RFT, the two-staged SFT-then-RFT baselines, other recent parallel works (Yan et al., 2025; Ma et al., 2025), and effectively expand the model's reasoning capability boundaries (evaluated by pass@2024 on AIME). The method is validated across different model scales, model families, and demonstration quantities and qualities. Our further analysis reveals that Prefix-RFT enables the model to solve problems where the RFT struggles and also pushes it to learn more from demonstrations for challenging problems than for easier ones. Taken together, our work reconsiders the view that treats SFT and RFT as two distinct and consecutive stages, suggesting that an integrated approach combining both learning paradigms could be a valuable direction.

**Conflict of Interest Disclosure**   One author is affiliated with Qwen Team, Alibaba Group. This paper includes experiments with publicly available Qwen-series models, evaluated under the same protocols as the other models.

## 2. A Unified View on SFT and RFT

In the mainstream LLM training pipeline, SFT and RFT are typically regarded as two distinct stages. This section demonstrates that, despite originating from different theoretical foundations, the core dynamics of their parameter updates are inherently consistent. For simplicity, we only consider the training for one data point. We use $t$ to denote the token index in that data point and use the $\pi_\theta$ to represent the model to optimize.

**SFT**   The SFT training objective seeks to imitate expert demonstrations from an offline expert policy $\pi_{\text{off}}$. Thus, for a model $\pi_\theta$, a prompt $x$, and a demonstration $y^* \sim \pi_{\text{off}}(\cdot|x)$, the SFT loss and its gradient are

$$L_{\text{SFT}}(\theta) = -\log \pi_\theta(y^*|x)$$
$$\Rightarrow \nabla_\theta L_{\text{SFT}} = -\sum_t \nabla_\theta \log \pi_\theta(y_t^*|x, y_{<t}^*) \quad (1)$$

The gradient $\nabla_\theta L_{\text{SFT}}$ provides a low-variance signal that pushes the model $\pi_\theta$ directly towards the expert data distribution $\pi_{\text{off}}$, which is usually *not directly accessible*.

**Policy Gradient**   On the other hand, RFT methods use the current policy $\pi_\theta$ to generate the rollout $y \sim \pi_\theta(\cdot|x)$ and collect rewards, then use these samples to compute the policy gradient to update the policy model $\pi_\theta$ (Sutton et al., 1999). Specifically, for LLM post-training, we can treat each token generation as a separate action. Thus, the policy gradient we use to update $\pi_\theta$ is

$$\nabla_\theta L_{\text{PG}} = \sum_t \hat{A}_t \nabla_\theta \log \pi_\theta(y_t|x, y_{<t})$$

where $\hat{A}_t$ is the estimated advantage for generating the token $y_t$, and is usually the same for all tokens in the response when applying value-model free RFT algorithms like GRPO (Shao et al., 2024). The two terms essentially decide *how much* and *in which direction* to update the policy.

**PPO Training Objective**   The vanilla policy gradient is strictly on-policy. To improve sample efficiency, RFT for LLMs generally employs the Proximal Policy Optimization (PPO) style objective (Schulman et al., 2017; Shao et al., 2024). The core idea is to enable multiple gradient updates with the collected samples, *i.e.*, the sample generated by the "old" policy $\pi_{\theta_{\text{old}}}(\cdot|x)$ is used to update the current policy $\pi_\theta(\cdot|x)$. Leveraging the importance sampling to correct the distribution shift and the clipping technique, the PPO training objective $L_{\text{PPO}}(\theta)$ can be expressed as:

$$L_{\text{PPO}}(\theta) = \sum_t \min\left[r_t \cdot \hat{A}_t, \text{clip}\left(r_t, 1-\epsilon, 1+\epsilon\right) \cdot \hat{A}_t\right]$$

where $r_t = \frac{\pi_\theta(y_t|x,y_{<t})}{\pi_{\theta_{\text{old}}}(y_t|x,y_{<t})}$ is the ratio between $\pi_\theta$ and $\pi_{\theta_{\text{old}}}$. Thus, its gradient is calculated as

$$\nabla_\theta L_{\text{PPO}} = \sum_t \mathbb{I}\Big(\{\hat{A}_t > 0 \wedge r_t \leq 1+\epsilon\}$$
$$\vee \{\hat{A}_t < 0 \wedge r_t \geq 1-\epsilon\}\Big)\hat{A}_t \nabla_\theta r_t(\theta)$$
$$= \sum_t \mathbb{I}_{\text{clip}}(r_t, \hat{A}_t)\hat{A}_t \nabla_\theta r_t$$
$$= \sum_t \mathbb{I}_{\text{clip}}(r_t, \hat{A}_t)\hat{A}_t r_t \nabla_\theta \log \pi_\theta(y_t|x, y_{<t})$$

Comparing $\nabla_\theta L_{\text{SFT}}$, $\nabla_\theta L_{\text{RFT-PG}}$ and $\nabla_\theta L_{\text{PPO}}$, all methods function by applying a gradient to the log probability of a

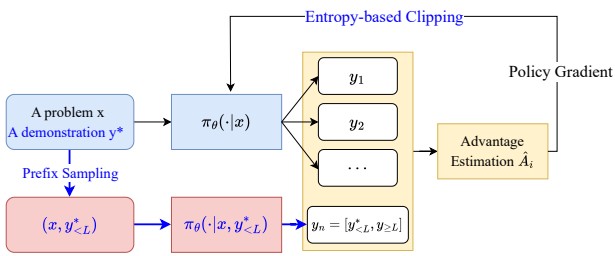

*Figure 1.* Given a problem and a demonstration, a prefix is sampled to guide the online continuation. The concatenated sequence $y_n$ is mixed with other online rollouts to perform RFT-style training.

sequence, where $\nabla_\theta L_{\text{SFT}}$ updates the log probability of an expert sequence $y^*$, implicitly treating its advantage as 1. $\nabla_\theta L_{\text{RFT-PG}}$, weighted by the advantage $\hat{A}_t$, leads the model to adjust the log probability of the trial-and-error-discovered trajectories. $\nabla_\theta L_{\text{PPO}}$ takes one step further. The gradient of the log probability is multiplied by a dynamic, per-token weight $\mathbb{I}_{\text{clip}}(r_t, \hat{A}_t)\hat{A}_t r_t$ where the clipping $\mathbb{I}_{\text{clip}}(r_t, \hat{A}_t)$ penalizes large policy changes.

**Hybrid Approach** Given this inherent consistency, we propose a hybrid post-training objective for blending SFT and RFT. Consider a set of $N$ responses $\{y^{(1)}, \ldots, y^{(N)}\}$ for a prompt $x$. These responses may originate entirely from the online policy $\pi_{\theta_{\text{old}}}$, the offline expert policy $\pi_{\text{off}}$, or a composition of both. For the $i$-th response, we partition the token indices into two sets: $\mathcal{T}_{\text{exp}}^{(i)}$ for tokens generated by the model (exploration) and $\mathcal{T}_{\text{imit}}^{(i)}$ for tokens from offline demonstrations (imitation). The gradient used to optimize the policy is formulated as:

$$\nabla_\theta L_{\text{Hybrid}} = -\frac{1}{N} \sum_{i=1}^{N} \underbrace{\sum_{t \in \mathcal{T}_{\text{exp}}^{(i)}} \alpha_{i,t} \nabla_\theta \log \pi_\theta(y_t^{(i)}|x, y_{<t}^{(i)})}_{\text{learning from exploration}}$$
$$-\frac{1}{N} \sum_{i=1}^{N} \underbrace{\sum_{t \in \mathcal{T}_{\text{imit}}^{(i)}} \beta_{i,t} \nabla_\theta \log \pi_\theta(y_t^{(i)}|x, y_{<t}^{(i)})}_{\text{learning from imitation}}$$

(2)

where $\alpha_{i,t}$ and $\beta_{i,t}$ represent the specific weights assigned to the $t$-th token of the $i$-th response.

## 3. Prefix Reinforcement Fine-Tuning

SFT and RFT paradigms demonstrate clear complementarity. Specifically, SFT pushes the model to fit the target distribution but may provide overly restrictive supervision, and sometimes its training signal aligns poorly with downstream task performance. However, it still provides reliable

optimization directions and ensures that the model captures accurate problem-solving patterns in high-quality data. On the contrary, RFT uses generations from the model itself to gently carve the model's behaviors. Although promising results have been achieved, recent research raises questions about whether it can truly lift the upper bound of the base policy, suggesting that the absence of explicit external guidance on exploration may severely limit how far we can go with pure RFT. By combining them into a hybrid fine-tuning framework, we aim to benefit from SFT's stable knowledge acquisition while leveraging reinforcement learning's exploratory power. As shown in Fig. 1, Prefix-RFT is to *use offline demonstration prefixes $y_{<L}^*$ as guiding hints and then mix prefixes with their on-policy continuation and other on-policy rollouts to perform RFT-style training*.

Given a prompt $x$ and a demonstration $y^*$, we start as in a standard RFT pipeline by generating $N-1$ online rollouts $\{y^{(1)}, \ldots, y^{(N-1)}\}$ with the current policy $\pi_{\theta_{\text{old}}}$. For the $N$-th sequence, we truncate $y^*$ to a prefix $y_{<L}^*$ and use $\pi_{\theta_{\text{old}}}$ to sample its continuation $y_{\geq L}$. We then stitch the $y_{<L}^*$ and $y_{\geq L}$ to form the hybrid trajectory $y^{(N)}$. This keeps the same rollout budget as standard RFT: one standard on-policy rollout is replaced by a prefix-guided hybrid rollout, rather than adding an extra trajectory. All sequences are used to estimate the advantages $\hat{A}_t$. We follow the unified view in Eq. 2 by setting the token-specific weights $\alpha_{i,t} = \beta_{i,t} = \mathbb{I}_{\text{clip}}(r_t, \hat{A}_t)\hat{A}_t r_t$, where $r_t$ is the standard PPO probability ratio. Consequently, the gradient update becomes:

$$-\frac{1}{N} \left( \underbrace{\sum_{i=1}^{N-1} \sum_t \mathcal{W}_{i,t}^{\text{PPO}} \nabla \log \pi + \sum_{t \geq L} \mathcal{W}_{N,t}^{\text{PPO}} \nabla \log \pi}_{\text{Exploration:Standard Rollouts + Continuation}} \right.$$
$$\left. + \underbrace{\sum_{t < L} \mathcal{W}_{N,t}^{\text{PPO}} \nabla \log \pi}_{\text{Imitation: Prefix Guidance}} \right)$$

(3)

where $\mathcal{W}_{i,t}^{\text{PPO}} = \mathbb{I}_{\text{clip}}(r_t, \hat{A}_t)\hat{A}_t r_t$ denotes the clipped PPO weight. This formulation highlights that while the prefix tokens ($t < L$) originate from the offline expert, they are reinforced using the advantage estimated from the full hybrid trajectory. The intuition behind prefix sampling is to provide the model with partial guidance, allowing it to explore the continuation rather than forcing it to mimic the entire sequence. Using the same ratio and clipping mechanism as in PPO, we penalize large updates from the demonstration data. Meanwhile, the advantage assigned can indicate the value of the given prefix. Therefore, for prompts where the model does not perform well, the high-quality prefix will receive higher gradient weights, enabling the model to benefit more from this partial demonstration.

**Entropy-based Clipping for Constrained Update on Demonstrations** In practice, $\pi_{\text{off}}$ may be far from the current policy, and the probability $\pi_\theta$ of offline tokens is therefore generally low. In this case, the gradient of demonstrations could be significantly larger than the RFT gradients (as illustrated in Table 4 in the App. A.3), potentially dominating the optimization process and preventing the model from learning via RFT. To this end, we propose an entropy-based clipping approach, *i.e.*, that involves only the top-k% high-entropy demonstration tokens. Regarding implementation, we directly set the corresponding advantages of all other tokens to zero, thereby removing their contribution to the gradient. Our principle is not that entropy identifies the uniquely correct demonstration tokens. Instead, entropy clipping acts as a conservative off-policy filter: low-entropy prefix tokens are often either already matched by the current policy, and thus contribute little learning signal, or confident mismatches that induce sharp overwrite updates. By retaining high-entropy prefix tokens, we target positions where the model is uncertain while still letting the trajectory-level advantage decide whether the prefix should be reinforced. We provide further token-level evidence and discussion in App. A.4.

**Controlling Prefix Length with Cosine Decay Scheduler** In practice, we use a variable $l \in [0, 1]$ to determine the prefix length as $L = \lfloor l \cdot |y^*| \rfloor$. If $l \sim U(0, 1)$, the model naturally has a higher chance of accessing early tokens in the demonstration. However, specific skills, such as drawing conclusions or summarizing, are usually located at the end of the sequence. To alleviate this positional bias, we propose using a cosine-decay scheduler to control the prefix length. Specifically, the length variable $l$ is randomly sampled from $U(low, high)$, where $high$ is a constant and $low$ decreases from $high$ to near zero throughout the entire training. The design not only mitigates the position-bias issue but also incorporates a curriculum learning schedule into training, naturally aligning with the existing standard SFT-and-then-RFT recipe.

**Related Works** Note that several parallel works share a similar motivation to incorporate the offline dataset into RFT training. LUFFY (Yan et al., 2025) mixes the entire offline data with other on-policy rollouts to perform RFT-style training. More similar to us, UFT (Liu et al., 2025b) first samples a prefix from the demonstration, then uses SFT loss on the prefix with a static small weight and RFT loss on the on-policy continuations. ReLIFT (Ma et al., 2025) incorporates a staged method to interleave SFT and RFT, with the SFT focusing on challenging problems that RFT cannot solve. Compared with these methods, our approach is distinguished by its practicability, simplicity, and ease of integration into existing RFT pipelines. We include these related works as a baseline, and further discussion of related

works and a detailed comparison with these parallel works can be found in App. A.1.

## 4. Main Experiments

**Experiment Settings and Baselines** We employ mathematical reasoning as the test playground due to access to reliable and inexpensive verifiers. Our training data is a length-filtered subset (Yan et al., 2025) of the OpenR1-Math-220K dataset (Face, 2025), comprising approximately 46k problems, each problem equipped with a demonstration generated by DeepSeek-R1. We mainly use Qwen2.5-Math-7B (Yang et al., 2024) as our base model. The evaluation and other training details are included in the Appendix A.2. We compare with the following baselines: (1) Previous RFT recipes, including *Simple-RL* (Zeng et al., 2025) and *Oat-Zero* (Liu et al., 2025d). (2) For a fair comparison, all the following baselines use the same base model and the same dataset. The difference solely lies in how to incorporate offline data points. These baselines include *RFT*, *SFT*, *RFT w/ SFT Loss* that directly employs SFT loss on the off-policy data during RFT training, *SFT + RL* that continues RFT training with the SFTed model. Recent hybrid approaches *ReLIFT*, *UFT*, and *LUFFY* are also included as our baselines.

**Main Results** The results are shown in the Tab. 1. Our observations and conclusions are as follows: **(1)** The pure *RFT* baseline in our setting already achieves strong performance compared to established Zero-RL results, validating that the setting and comparison of our experiments are solid. **(2)** In our setting, the *RFT* and *SFT* baselines achieve similar performance, while *SFT+RFT* is significantly better. In particular, *SFT* contributes more on more challenging benchmarks (*i.e.* AIME25), *RFT* benefits more on datasets where the base model already has moderate performance (*i.e.* AMC and MATH500), and *SFT+RFT* achieves a better balance, highlighting our motivation that those two learning paradigms could complement each other and could be better blended. **(3)** The joint training method *RL w/ SFT Loss* is counterproductive, validating our concerns about gradient dominance when learning from both losses directly. **(4)** Though utilizing less demonstration data, the multi-staged method like *ReLIFT* does not necessarily achieve better performance than the simple two-staged *SFT+RFT* baseline. This may be because the interleaved method requires more careful hyperparameter tuning to ensure the performance. This also highlights the need for a hybrid approach to stably exploit the offline dataset during RFT. **(5)** Despite its simplicity, Prefix-RFT performs well on six math reasoning benchmarks and three general domain reasoning tasks, significantly outperforming all baselines and achieving comparable performance with concurrent work *LUFFY* (we provide significance tests in Tab. 5 in App. A.3).

*Table 1.* Main experiment results on math and general reasoning benchmarks based on **Qwen2.5-Math-7B**.

| Model | Math Reasoning Performance | | | | | | General Domain Reasoning Performance | | | |
|---|---|---|---|---|---|---|---|---|---|---|
| | AIME 24/25 | AMC | MATH-500 | Minerva | Olympiad | Avg. | ARC-c | GPQA* | MMLU-Pro | Avg. |
| Qwen2.5-Math-7B | 11.5/4.9 | 31.3 | 43.6 | 7.4 | 15.6 | 19.0 | 18.2 | 11.1 | 16.9 | 15.4 |
| *Previous RFT Results* | | | | | | | | | | |
| SimpleRL-Zero | 27.0/6.8 | 54.9 | 76.0 | 25.0 | 34.7 | 37.4 | 30.2 | 23.2 | 34.5 | 29.3 |
| Oat-Zero | **33.4**/11.9 | 61.2 | 78.0 | 34.6 | 43.4 | 43.7 | 70.1 | 23.7 | 41.7 | 45.2 |
| *Baselines Using the Same Dataset and Base Model* | | | | | | | | | | |
| RFT | 25.1/15.3 | 62.0 | 84.4 | 39.3 | 46.8 | 45.5 | 82.3 | **40.4** | 49.3 | 57.3 |
| SFT | 22.2/22.3 | 52.8 | 82.6 | **40.8** | 43.7 | 44.1 | 75.2 | 24.7 | 42.7 | 47.5 |
| RL w/ SFT Loss | 19.5/16.4 | 49.7 | 80.4 | 34.9 | 39.4 | 40.1 | 71.2 | 23.7 | 43.2 | 46.0 |
| SFT+RFT | 25.8/23.1 | 62.7 | 87.2 | 39.7 | 50.4 | 48.2 | 72.4 | 24.2 | 37.7 | 44.8 |
| UFT | 20.8/16.5 | 58.8 | 83.8 | 33.8 | 51.6 | 44.2 | 83.4 | 34.5 | 49.4 | 55.8 |
| ReLIFT | 28.2/20.1 | 64.9 | 87.4 | 33.8 | 52.5 | 47.8 | 76.2 | 37.9 | 52.5 | 55.5 |
| LUFFY | 29.4/23.1 | 65.6 | 87.6 | 37.5 | **57.2** | 50.1 | 80.5 | 39.9 | **53.0** | 57.8 |
| *Our Method* | | | | | | | | | | |
| Prefix-RFT | 31.8/**26.4** | **68.2** | **88.4** | 40.3 | 55.7 | **51.8** | **84.0** | 39.1 | 52.1 | **58.4** |

**Results on more models** We also experiment with Qwen2.5-Math-1.5B, LLaMA-3.1-8B, and Qwen3-base-1.7B models. Detailed results can be found in the Tab. 6, App. A.3. The results clearly indicate the superior performance of our method. On the Qwen2.5-Math-1.5B model, Prefix-RFT achieved an average score of 41.1, significantly outperforming the next-best method, LUFFY, as well as the conventional SFT and RFT methods. A similar trend was observed on the LLaMA-3.1-8B and Qwen3-1.7B-base models, highlighting its robustness.

**Pass@k results** To investigate whether Prefix-RFT expands the model's reasoning boundaries beyond simply improving the likelihood of known solutions, we evaluate the model's performance with a large sampling budget ($n = 2048$). The results are visualized in Fig. 2 and listed in Tab. 7, revealing several key insights. First, RFT's performance at $k = 2048$ remains remarkably close to the base model, suggesting that it struggles to effectively elevate the model's upper bound. Second, SFT leads to a performance degradation at larger sampling budgets, indicating that it may overly constrain the model's solution space and reduce the diversity necessary for successful high-budget sampling. Furthermore, despite incorporating offline demonstrations, LUFFY fails to significantly push the performance ceiling beyond that of RFT. In contrast, Prefix-RFT consistently outperforms all baselines across the entire sampling spectrum, and is the only method that successfully raises the performance upper bound, achieving a consistent 6.67 percentage point improvement in Pass@2048 over the base model on both the AIME24 and AIME25.

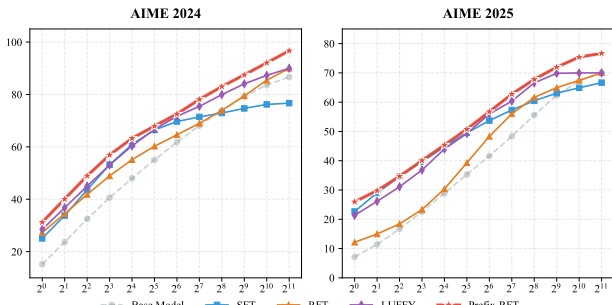

*Figure 2.* Pass@k results on AIME2024 and AIME2025 datasets.

## 5. Analysis

To better validate the underlying mechanisms of our proposed method, we investigate the following two questions: (1) *Does Prefix-RFT effectively synthesize the distinct training paradigms of SFT and RFT?* (2) *Does Prefix-RFT dynamically adjust its learning strategy during different training stages and when faced with problems of varying difficulty?* The detailed settings for our analysis experiments are as follows. Most analysis experiments are based on Qwen2.5-Math-1.5B. We subsample a 16k dataset $\text{Train}_{16k}$ from the original training dataset. We use a batch size of 128 and train 5 epochs. The learning rate is set as 5e-5 for SFT and 1e-6 for RFT and Prefix-RFT . All different analysis metrics are calculated with a 2k subset from $\text{Train}_{16k}$, noted as $\text{Train}_{2k}$. To study the model's behavior on problems that the pure RFT struggles to solve, we use checkpoints from

another RFT run to identify 256 problems and note this subset as $\text{Train}_{\text{hard}}$.

## 5.1. Does Prefix-RFT bridge SFT and RFT?

To answer this question, we calculate three metrics to measure the model's state during the training with $\text{Train}_{2k}$: (1) the Avg@16 score, which estimates the model's task performance; (2) the Best@16 score, which estimates the model's problem-solving potential after training; and (3) the SFT loss on the provided demonstrations that estimate the distribution gaps between the model and the offline expert policy $\pi_{\text{off}}$. These results are summarized in Fig. 3. Our key observations are as follows.

**Comparing SFT and RFT** As mentioned above, SFT and RFT present distinct training paradigms: the former focuses on minimizing the likelihood of the given demonstration, while the latter directly optimizes for task performance. Our results suggest that (1) The SFT could initially damage the model's performance and then rebuild it (the performance of the second checkpoint is lower than that of the initial one). Mimicking external demonstrations, the SFT-ed model can find the solution path for nearly all training problems (higher best@16 of 0.96), but cannot robustly solve them (lower avg@16 of 0.52). (2) On the contrary, our results indicate that the RFT model achieves better overall performance (0.61 for RFT v.s. 0.52 for SFT regarding avg@16) but can be limited by the ability of the initial policy (0.85 for SFT v.s. 0.14 for RFT regarding best@16 on $\text{Train}_{\text{hard}}$). Furthermore, the best@16 converges much faster than avg@16, implying RFT admits a discover-and-gradually-reinforce learning strategy. (3) During the RFT training, the model keeps deviating from the demonstration distribution (loss on demonstration increased from 0.67 to 0.81), proving that SFT's training objective—loss on demonstration—can sometimes be a poor predictor of the exact task performance. In total, all these observations reflect the pros and cons of the two methods and their complementarity, justifying our motivation for blending both learning paradigms.

**Prefix-RFT makes the best of both worlds (to some extent)** Our results demonstrate the superiority of Prefix-RFT . Compared to RFT, Prefix-RFT not only achieves higher avg@16 and best@16, and lower loss on demonstration, but also shows notable performance gains on problems previously intractable for RFT. Notably, our approach effectively aligns with the offline expert distribution despite fine-tuning only on the top 20% of high-entropy tokens. We posit that the observed gap in final loss between Prefix-RFT (0.45) and SFT (0.35) is likely attributable to hyperparameter settings rather than fundamental methodological differences. To investigate this, we conducted preliminary tests on hyperparameter sensitivity. For instance, training SFT with a lower learning rate of 1e-6 results in a final loss of ap-

proximately 0.42. Conversely, applying a higher learning rate of 5e-5 to RFT leads to training instability and eventual model collapse. These findings underscore that identifying optimal hyperparameters presents a significant challenge for unified training frameworks, which we designate as an avenue for future work. Furthermore, a performance gap persists between Prefix-RFT and SFT on the $\text{Train}_{\text{hard}}$ dataset, suggesting room for improvement.

## 5.2. Advantage-driven updates induce dynamic transition between SFT and RFT

As discussed in the Sec. 3, the advantage assigned to a hybrid sequence serves as a proxy for its prefix's utility. This advantage dynamically adjusts the prefix's influence on the training update, thereby inducing a transition between SFT and RFT. We find that this transition manifests at both the level of overall training dynamics and individual examples.

**Overall training dynamics** Fig. 4 plots the average reward of rollouts initiated with a prefix and the overall training reward of Qwen2.5-Math-1.5B and Qwen2.5-Math-7B. According to the advantage definition in GRPO algorithms, the shaded area between these two curves roughly represents the accumulated advantage assigned to the prefix. Here are our observations: (1) Both models demonstrate a remarkable ability to quickly leverage the provided prefix. The "Reward with Prefix" score surges in the initial phase, with the 7B model approaching a near-perfect score of 1.0 within the first 100 training steps. (2) Owing to the cosine decay scheduler, the average reward with the prefix slightly decreases. The behavior is clearer for the 1.5B model, suggesting that the smaller model is more sensitive to changes in prefix length. (3) The gap between the two average rewards stays positive and gradually narrows down through the training. This diminishing advantage signifies that the model's reliance on the prefix decreases as its own generative reasoning capabilities improve. *From the perspective of the gradient, this trend reflects a smooth and desirable transition from SFT to RFT during the training process.*

**Individual example level** We then investigate whether such a transition also exists at the example level. We analyze changes in SFT loss on demonstrations across problems of varying difficulty. This analysis focuses on the training interval from the first to the third epochs, a phase chosen to bypass the initial rapid convergence and subsequent performance saturation in later stages. Figure 5 presents the primary results of this analysis. Each point in the scatter plot corresponds to a unique data point, plotting its difficulty against the observed change in SFT loss. Problem difficulty (x-axis) is quantified as the model's mean solution accuracy across multiple saved checkpoints from the 2nd to the 3rd epochs. A lower accuracy indicates a more

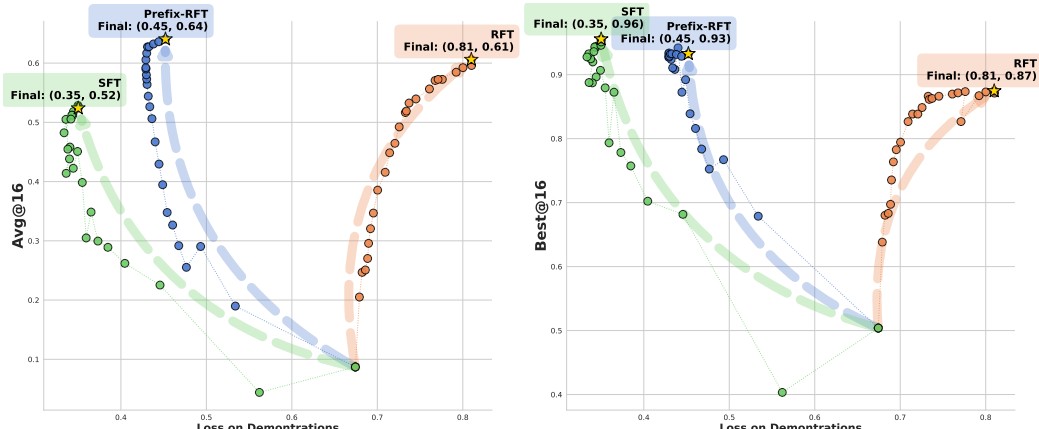

*(a)* Training trajectories on the Train$_{2k}$. The figure highlights the distinct learning objectives and paradigms of SFT and RFT, and indicates that Prefix-RFT effectively blends both methods regarding training objectives.

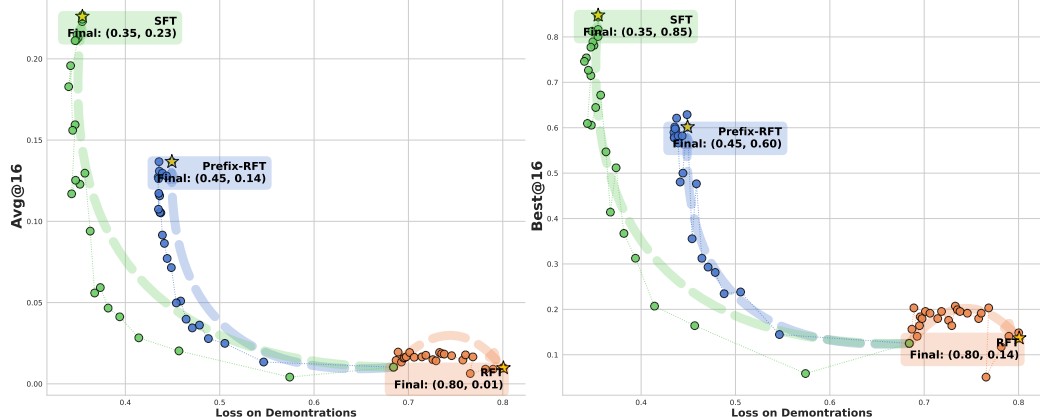

*(b)* Training trajectories on the Train$_{hard}$. It shows that the RFT method continues to struggle with these unsolvable problems during training, and Prefix-RFT achieves higher scores, effectively raising the upper bound of RFT tuning.

*Figure 3.* Training trajectories of SFT, RFT, and Prefix-RFT. The x-axis denotes the SFT loss on the demonstrations. The y-axis represents the Avg@16 and Best@16 scores. The final step is marked with a yellow star and annotated with the final SFT loss and score.

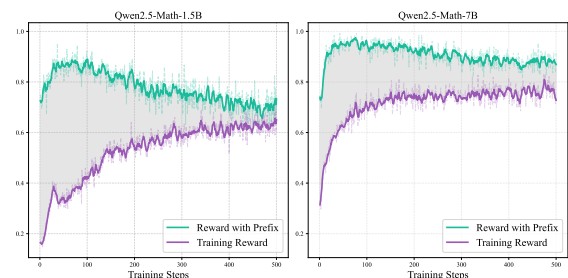

*Figure 4.* The rewards of rollouts with a prefix and the overall rewards. The decreasing shaded area represents the diminishing advantage assigned to the prefix, indicating that the training gradually transitions from SFT to RFT.

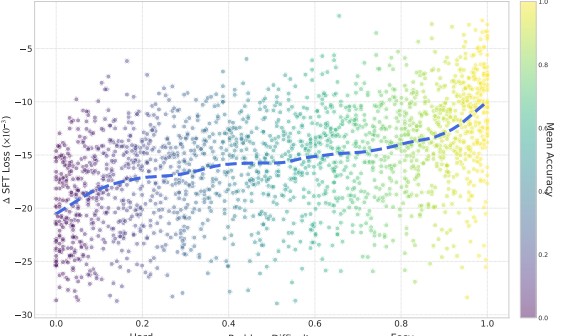

*Figure 5.* The change in SFT loss on problems of varying difficulties, suggesting our method provides more supervision for more challenging problems.

challenging problem. The y-axis represents the change in SFT loss on the provided demonstrations. A more negative value signifies more learning pressure from demonstration loss. The LOWESS-fitted trend line reveals a clear positive correlation: as mean accuracy increases, the change in SFT loss becomes less negative. This indicates that the

model achieves substantially greater loss reduction—and thus learns more intensively from the demonstrations—for problems it finds more challenging (i.e., those with lower accuracy). Conversely, for easier problems where the model

already achieves high accuracy, the SFT loss reduction is marginal. This suggests that the model relies less on the demonstrations and more on its own problem-solving abilities. *This finding elucidates a key mechanism of our approach: it facilitates a dynamic, example-level transition between reliance on demonstrations and self-exploration.*

## 6. Ablation Studies

**Entropy-based Clipping**    To validate the effect of entropy-based clipping, we compare five different approaches. Our primary method, labeled *top 20%*, updates the model using only the 20% of prefix tokens with the highest entropy. We compare this against four variants. Two of these, *top 50%* and *top 80%*, maintain the high-entropy selection strategy but relax the clipping ratio to 50% and 80%, respectively. The other two variants serve as controls: *random 20%* selects tokens at random, while *bottom 20%* selects the 20% of tokens with the lowest entropy. All variants were trained for 300 steps, during which we monitored training response length, training reward, and benchmark performance.

The results clearly demonstrate the superiority of the top 20% strategy. Regarding benchmark performance, the top 20% variant exhibits a stable upward trend, achieving the highest score of approximately 50% after 300 steps and the highest training reward. Its superior performance was achieved while generating the shortest training response length (2k–2.5k tokens). In contrast, relaxing the clipping ratio ( top 50% or 80%) or altering the selection method (random or bottom) leads to diminished performance and greater instability. These results confirm our core motivations about the clipping strategy: (1) *The update on demonstrations should be constrained*: When an offline dataset is significantly off-policy, the gradients from the prefix tokens can overwhelm those from on-policy tokens. This can cause the training to degenerate into simple SFT on the prefix data. This phenomenon is evident in the top 80% variant, which quickly overfits to the superficial feature of response length from the demonstrations rather than optimizing for task performance. This finding aligns with our preliminary experiments on using multiple prefixes (sampled from the same demonstration) for a single problem (similar to UFT). The hybrid approach becomes counterproductive in this case, as the model struggles to balance both learning signals. (2) *High-entropy tokens provide richer learning signal*: Merely constraining the update ratio is insufficient; the strategy for selecting tokens is crucial. As shown in the figure, the *random 20%* is only a delaying tactic and is ineffective at preventing the policy from overfitting to the demonstrations. The *bottom 20%* strategy was even worse, proving detrimental to both training reward and benchmark performance. This confirms that focusing on high-entropy tokens is essential for effective learning.

**Decay Scheduler**    To investigate the effect of the proposed cosine decay scheduler, we conduct an ablation study comparing it against a *Uniform Scheduler* baseline: sampling the variable $l$ uniformly from the $[0.05, 0.95]$ throughout the entire training. The experiments are performed on both the Qwen2.5-Math-7B and 1.5B models. As illustrated in Fig. 6b (left), *Uniform Scheduler* incurs a performance degradation, albeit while still remaining superior to the naive SFT and RFT baselines. Beyond mitigating potential positional bias, our proposed cosine decay scheduler also more effectively modulates the training dynamics. As shown in Fig. 6b (right), under the uniform schedule, the gap between the prefix-initiated and the overall reward is initially small and gradually widens, which is in contrast to the pattern observed in Fig. 4, suggesting that our cosine decay scheduler better incentivizes the model to learn from the demonstration, particularly during the early training stage.

## 7. Additional Results and Discussions

**Data efficiency and demonstration quality**    Compared with RFT, Prefix-RFT requires extra demonstrations. Since acquiring such data can be prohibitively expensive, either from human experts or a superior model, we investigate the method's performance under two data-constrained scenarios to assess its real-world viability: (1) limited demonstration quantity, using 10% (4.5k) and 1% (0.45k) of the training data; and (2) suboptimal demonstration quality, with demonstrations generated by DeepSeek-R1 distillation series models of varying sizes (1.5B to 32B). All experiments use the Qwen2.5-Math-1.5B model, with results detailed in Table 2. The analysis shows that even under these strict constraints, all variants of Prefix-RFT still significantly outperform the SFT and RFT baselines. Regarding data quantity, reducing the training set by 99% (from 45k to 0.45k samples) results in only a moderate performance drop (40.8 to 37.6), highlighting its data efficiency. The method also shows remarkable robustness to demonstration quality, as performance is nearly identical when using a 1.5B generator versus a 32B one. We note, however, that the most challenging benchmarks, such as AIME, are the most affected by these limitations, indicating that high-quality, large-scale demonstration data remains beneficial for tackling top-difficulty problems.

**Code generation**    To further test whether Prefix-RFT is tied to mathematical reasoning, we conduct a preliminary code-generation experiment with Qwen3-1.7B-Base. We use Coder1 training data with GPT-5.4-generated demonstrations, and evaluate on MBPP+, HumanEval+, and the Coder1 test split using the same training recipe as in our main experiments. As shown in Table 3, Prefix-RFT consistently improves over RFT on all three code benchmarks. This supports that the core design is not specific to math:

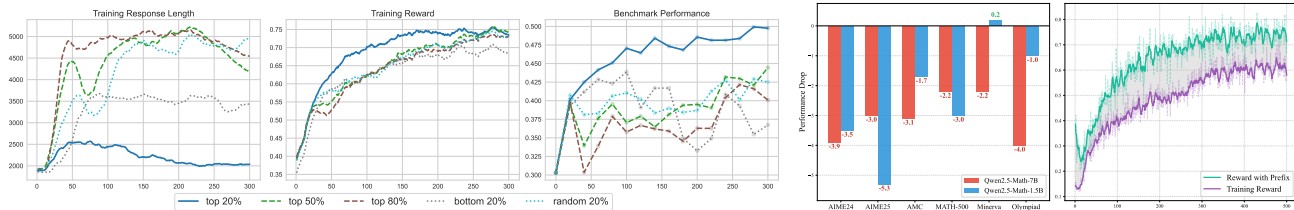

*(a)* Ablation study on entropy-based clipping strategy.  *(b)* Effect of the scheduling strategy.

*Figure 6.* Ablation studies on key components. (a) shows training length, reward, and benchmark performance for the entropy strategy. (b) compares the cosine decay scheduler vs. the uniform scheduler and training reward dynamics.

*Table 2.* Ablation study for Prefix-RFT on the Qwen2.5-Math-1.5B model. Results show that Prefix-RFT substantially outperforms SFT and RFT baselines while demonstrating strong data efficiency and remarkable robustness to the quality of demonstration generators.

|  | AIME 24/25 | AMC | MATH-500 | Minerva | Olympiad | Avg. |
|---|---|---|---|---|---|---|
| SFT | 11.7/13.2 | 37.8 | 70.6 | 26.8 | 31.3 | 31.9 |
| RFT | 11.8/7.7 | 40.2 | 61.8 | 26.8 | 32.0 | 30.0 |
| Prefix-RFT | 17.7/17.1 | 50.5 | 81.4 | 32.7 | 46.5 | 41.1 |
| *Ablations on different demonstration sizes* | | | | | | |
| Data size 4.5k | 17.8/15.9 | 49.7 | 79.0 | 35.3 | 46.8 | 40.8 |
| Data size 0.45k | 15.2/11.8 | 46.3 | 76.0 | 33.5 | 42.8 | 37.6 |
| *Ablations on different demonstration generators* | | | | | | |
| DeepSeek-R1-Distill-32B | 18.1/15.3 | 50.9 | 81.2 | 34.2 | 43.7 | 40.6 |
| DeepSeek-R1-Distill-7B | 18.1/15.9 | 49 | 79.8 | 36.4 | 44.9 | 40.7 |
| DeepSeek-R1-Distill-1.5B | 15.9/12.6 | 47.7 | 79 | 37.1 | 46.2 | 39.8 |

*Table 3.* Code-generation results with Qwen3-1.7B-Base.

| Method | MBPP+ | HumanEval+ | Coder1-test |
|---|---|---|---|
| Base | 0.359 | 0.378 | 0.108 |
| SFT | 0.063 | 0.044 | 0.007 |
| SFT+RFT | 0.500 | 0.447 | 0.316 |
| RFT | 0.613 | 0.649 | 0.404 |
| Prefix-RFT | **0.640** | **0.689** | **0.429** |

the prefix is sampled by truncating a demonstration, and the main task-specific component is the outcome verifier.

**Additional ablations** Additional methodological ablations are in App. A.3. Learning from the prefix loss is crucial, as merely using prefixes as guidance (e.g., in DR-PO style) severely degrades performance to the standard RFT level (Table 8). Our dynamic advantage-based weighting also significantly outperforms static weighting, and the observed performance gain is indeed from explicitly reinforcing high-quality prefixes rather than simple entropy-clipping mechanisms (Table 8). Further discussion of the proposed two components and their corresponding hyperparameters is in App. A.4.

## 8. Conclusion

Motivated by the complementarity of SFT and RFT, this work presents Prefix-RFT to blend them via sampling prefixes from offline demonstrations as hints. The method is validated across different model scales and families, and with varying demonstration quantities and qualities, to demonstrate its robustness. Our design choices are justified with extensive ablation studies. Further analysis highlights that Prefix-RFT effectively guides the model to solve problems that are unlearnable to pure RFT, striking a sweet spot between RFT (by providing supervision where it's most needed) and SFT (by incorporating a goal-oriented training objective). **Limitations:** Our experiments primarily focus on verifiable reasoning settings in which outcome rewards can be obtained reliably. While additional code-generation results suggest that the same recipe can transfer to another verifier-based domain, broader open-ended generation and noisy-reward settings remain important future work. Prefix-RFT also relies on external demonstrations; when multiple candidate demonstrations are available for each prompt, a natural strategy is to sample among them and let the trajectory-level advantage weight their utility, but systematic demonstration selection is left for future study.

## Acknowledgements

IT acknowledges support from Dutch National Science Foundation (NWO Vici grant VI.C.212.053). EP is supported by the ERC Starting Grant AToM-FM (101222956).

## Impact Statement

This paper presents work aimed at advancing the field of LLM Post-training. There are many potential societal consequences of our work, none of which we feel must be specifically highlighted here.

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

# A. Appendix

## A.1. Related Works

**Incorporating offline dataset for online RL**    Offline RL aims to learning a reward maximizing policy from a fixed, static dataset, collected by some existing policy (Levine et al., 2020; Lange et al., 2012). Due to the dataset limitations, offline RL often results in a suboptimal policy, motivating recent work to combine offline and online RL (Luo et al., 2023; Ball et al., 2023; Song et al., 2023; Liu et al., 2025c). In the domain of Large Language Model, the most common approach is to employ the two-stage SFT-and-then-RFT method, where SFT instills desirable patterns or skills into the model and RFT amplifies them (Liu et al., 2025a). Built upon this two-staged strategy, recent works like DR-PO (Chang et al., 2024) also investigates starting online RL from a sampled offline state (a truncated sentence for LLM) for RLHF. However, the interplay between SFT and RFT remains to be understood and may be specific to each use case (Cai et al., 2025; Chen et al., 2025a), and determining the optimal strategy to stitch the two methods remains an open question (Chen et al., 2025b). Therefore, there is an emerging body of work focusing on how to better integrate these two learning paradigms and how to incorporate the offline dataset to improve the LLM post-training (Yan et al., 2025; Liu et al., 2025b; Ma et al., 2025).

**Detailed comparison with parallel works**    Here we discuss the differences between our method and the contemporaneous works UFT, LUFFY, and ReLIFT. (1) **UFT** (Liu et al., 2025b) shares the strategy of sampling prefixes; it applies a static, pre-assigned scalar weight (0.001) to the prefix tokens. Our method, conversely, employs a dynamic weight determined by the estimated advantage of the full hybrid sequence. This allows the model to learn more intensively from prefixes that actually lead to high-value outcomes. Though theoretically justified, UFT mainly experiments with simpler settings, *i.e.*, smaller models and simpler offline demonstrations. Our empirical results demonstrate that our dynamic approach significantly outperforms the static weighting scheme in complex reasoning tasks. (2) **LUFFY** (Yan et al., 2025) mixes entire offline traces with on-policy data and utilizes a policy reshaping function $f(\pi_\theta) = \frac{\pi_\theta}{\pi_\theta + \lambda}$ to handle off-policy distribution shift. This method introduces a hyperparameter $\lambda$ that can be difficult to tune and sensitive. Our approach addresses the off-policy nature of demonstrations through a simpler entropy-based clipping mechanism, which we show to be robust and effective across different models and scales. (3) **ReLIFT** (Ma et al., 2025) adopts a staged method to interleave SFT and RFT, where SFT is selectively applied to problems that RFT fails to solve. While effective, this requires managing a multi-stage pipeline and identifying "hard" problems iteratively. In contrast, our method seamlessly blends SFT and RFT in a single training stage by dynamically using prefixes to guide exploration on difficult problems, offering a simpler, more integrated pipeline.

## A.2. Evaluation and experiment hyperparameters

Regarding **evaluation**, we follow Yan et al. (2025) and evaluate our approach with six math reasoning tasks, *i.e.*, AIME 2024, AIME 2025 (Li et al., 2024), AMC (He et al., 2024), Minerva (Lewkowycz et al., 2022), OlympiadBench (He et al., 2024), and MATH-500 (Hendrycks et al., 2021). Because AIME 2024, AIME 2025, and AMC have fewer data points, we report avg@32, and for the other three benchmarks, we report pass@1. To see whether the reasoning learned can be generalized to other general reasoning problems, we test the model with ARC-c (Clark et al., 2018), GPQA-diamond (Rein et al., 2023), and MMLU-Pro (Wang et al., 2024).

**Hyperparameters**    Most of our training hyperparameters are set as (Yan et al., 2025) to ensure fair comparison. Regarding our method-specific hyperparameters, unless specified otherwise, we sample 8 rollouts per prompt, and one of them starts with the sampled prefix. And for each mini-batch, we only update the top 20% prefix tokens that are high-entropy. The model is trained for 500 steps. And each time step $t$, we sample $l$ uniformly from $[\text{low}_t, 0.95]$ to decide the prefix length as $l$ times the total demonstration length. And $\text{low}_t$ follows a cosine decay scheduler, starting from 0.95 and decaying to 0.05 at the 500th step. We use Dr.GRPO (Liu et al., 2025d) as our RFT algorithm.

## A.3. More experiment results

**Gradient Magnitude Analysis**    To justify the motivation of our entropy-based clipping strategy, we analyze the gradient magnitudes derived from off-policy demonstration tokens compared to on-policy RFT tokens. As discussed in Sec. 3, because the offline expert policy $\pi_{\text{off}}$ (e.g., DeepSeek-R1) may be distributionally distant from the current policy $\pi_\theta$, the model often assigns low probabilities to demonstration tokens, resulting in large gradients for the log-likelihood objective. We compare the average gradient norms of four training methods on Qwen2.5-Math-7B: (1) **SFT-R1-trace** (pure SFT on

the demonstrations), (2) **Prefix-RFT (update all)** which updates all prefix tokens without clipping, (3) **Prefix-RFT (freeze prefix)** which freezes the prefix (gradient = 0), and (4) **Standard RFT**. As shown in Table 4, the gradients from the full SFT trace are orders of magnitude larger than RFT gradients. More importantly, *Prefix-RFT (update all)* exhibits gradient norms nearly double those of *Prefix-RFT (freeze prefix)* and *RFT*, despite prefix tokens constituting only a small fraction $(5\% - 10\%)$ of the total tokens in the batch. This confirms that a small number of off-policy tokens can disproportionately dominate the optimization landscape if left unconstrained, leading to instability (e.g., response length explosion) as observed in our ablation studies.

*Table 4.* Comparison of gradient norms across different training stages. Results show that unclipped off-policy prefixes generate significantly larger gradients than on-policy rollouts.

| Method | 0-25 steps | 25-50 steps | 50-100 steps | 100-200 steps |
|---|---|---|---|---|
| SFT (R1 Traces) | 5.31 | 2.01 | 1.61 | 1.78 |
| Prefix-RFT (update all) | 0.44 | 0.27 | 0.27 | 0.27 |
| Prefix-RFT (freeze prefix) | 0.23 | 0.13 | 0.13 | 0.13 |
| RFT | 0.18 | 0.12 | 0.12 | 0.12 |

**Statistical Significance Analysis** To assess the robustness of our reported improvements and ensure they are not artifacts of random seed selection, we conducted 5 independent inference runs for both Prefix-RFT and the strongest baseline, LUFFY (checkpoint released from their paper). We report the mean and standard deviation across these runs in Table 5. The results demonstrate that Prefix-RFT consistently outperforms LUFFY across all benchmarks with tight confidence intervals. For instance, on AIME 2025, we observe a substantial margin of +4.07% (25.13 vs 21.06), confirming the statistical significance of our contribution.

*Table 5.* Comparing Prefix-RFT against the strongest concurrent baseline, LUFFY, with multiple inference runs.

| Method | MATH | Olympiad | Minerva | AIME 2024 | AIME 2025 | AMC | Avg. |
|---|---|---|---|---|---|---|---|
| LUFFY | $87.16 \pm 0.83$ | $54.28 \pm 0.51$ | $38.09 \pm 1.74$ | $28.46 \pm 0.45$ | $21.06 \pm 0.78$ | $66.17 \pm 0.41$ | 49.20 |
| **Prefix-RFT** | **$87.60 \pm 0.58$** | **$56.33 \pm 0.52$** | **$39.97 \pm 0.45$** | **$31.88 \pm 0.41$** | **$25.13 \pm 0.96$** | **$68.18 \pm 0.11$** | **51.52** |

**Results on more models** In addition to Qwen2.5-Math-7B, we also test our method on Qwen2.5-Math-1.5B, LLaMA-3.1-8B, and Qwen3-1.7B-Base. The training settings for LLaMA models follow exactly Yan et al. (2025). Our baselines include SFT, RFT, LUFFY, and ReLIFT. The performance on the six math reasoning problems is summarized in Tab. 6. The results clearly indicate that the Prefix-RFT achieves superior performance on both models, regardless of their distinct architectures, scales, and initial capabilities. On the Qwen2.5-Math-1.5B model, Prefix-RFT achieved an average score of 41.1, significantly outperforming the next-best method, LUFFY, as well as the conventional SFT and RFT methods. A similar trend was observed on the LLaMA-3.1-8B and Qwen3-1.7B-base models, validating its effectiveness and robustness.

**Performance with Large Sampling Budgets** To investigate whether Prefix-RFT expands the model's reasoning boundaries beyond simply improving the likelihood of known solutions, we evaluate the model's performance with a large sampling budget ($n = 2048$). We compare the Base model (Qwen2.5-Math-7B-Base), SFT (checkpoint from LUFFY paper), RFT, and Prefix-RFT on the challenging AIME 2024 and AIME 2025 benchmarks. As shown in Table 7, Prefix-RFT consistently outperforms both SFT and RFT across all sampling budgets ($k = 1$ to $k = 2048$). Notably, on AIME 2025, Prefix-RFT achieves a Pass@2048 of 76.7%, significantly higher than the Base model (70.0%) and RFT (70.0%), demonstrating that our method effectively elevates the upper bound of the model's reasoning capabilities.

**More Ablation Studies** We further investigate the specific contributions of our design choices by comparing Prefix-RFT with several variants. We categorize these ablations into two groups: (1) *Prefix Gradient Strategies* and (2) *Methodological Variants*. All experiments are conducted on Qwen2.5-Math-7B. We aim to answer the following questions?

- **Do we need to update the prefix?** We compare *Freeze Prefix* (gradients from prefix tokens are all clipped) and *Update All* (no clipping on prefix). As shown in Table 8, freezing the prefix results in performance similar to standard RFT

*Table 6.* Performance on Qwen2.5-Math-1.5B, LLaMA-3.1-8B and Qwen3-1.7B-base models.

| Model | AIME 24/25 | AMC | MATH-500 | Minerva | Olympiad | Avg. |
|---|---|---|---|---|---|---|
| **Qwen2.5-Math-1.5B-Base** | | | | | | |
| SFT | 11.7/13.2 | 37.8 | 70.6 | 26.8 | 31.3 | 31.9 |
| RFT | 11.8/7.7 | 40.2 | 61.8 | 26.8 | 32.0 | 30.0 |
| ReLIFT | 14.3/10.0 | 40.9 | 76.4 | 25.2 | 39.6 | 34.4 |
| LUFFY | 16.0/13.1 | 47.1 | 80.2 | 30.5 | 41.0 | 38.0 |
| Prefix-RFT | **17.7/17.7** | **50.5** | **81.4** | **32.7** | **46.5** | **41.1** |
| **LLaMA-3.1-8B-Base** | | | | | | |
| SFT | 0.5/0.1 | 5.4 | 20.2 | 4.0 | 5.3 | 5.9 |
| LUFFY | **1.9**/0.1 | **13.5** | 39.0 | 15.1 | 9.6 | 13.2 |
| ReLIFT | 1.3/0.2 | 11.9 | 35.2 | - | 11.0 | - |
| Prefix-RFT | 1.3/**1.5** | 13.3 | **40.6** | **18.1** | **11.9** | **14.5** |
| **Qwen3-1.7B-Base** | | | | | | |
| SFT | 10.0/**10.4** | 34.5 | 66.6 | 24.6 | 27.2 | 28.9 |
| RFT | **11.6**/4.5 | 35.4 | 68.4 | 30.1 | 31.9 | 30.3 |
| Prefix-RFT | 10.0/8.1 | **36.3** | **74.2** | **32.7** | **35.3** | **32.8** |

*Table 7.* Pass@$k$ performance on AIME 2024 and AIME 2025 with a sampling budget of $n = 2048$.

| Method | p@1 | p@2 | p@4 | p@8 | p@16 | p@32 | p@64 | p@128 | p@256 | p@512 | p@1024 | p@2048 |
|---|---|---|---|---|---|---|---|---|---|---|---|---|
| **AIME 2024** | | | | | | | | | | | | |
| base model | 15.22 | 23.62 | 32.49 | 40.59 | 48.00 | 55.03 | 61.79 | 68.11 | 73.88 | 79.21 | 83.59 | 86.67 |
| SFT | 25.05 | 33.78 | 43.60 | 53.18 | 60.95 | 66.45 | 69.67 | 71.43 | 72.93 | 74.67 | 76.20 | 76.67 |
| RFT | 27.29 | 34.42 | 41.83 | 48.91 | 55.11 | 60.23 | 64.59 | 68.98 | 73.90 | 79.48 | 85.42 | **90.00** |
| LUFFY | 28.47 | 36.75 | 45.01 | 53.06 | 60.39 | 66.69 | 71.64 | 75.47 | 79.92 | 84.05 | 87.28 | 90.00 |
| **Prefix-RFT** | 31.24 | 40.08 | 48.93 | 56.90 | 63.24 | 67.94 | 72.55 | 78.11 | 83.01 | 87.48 | 92.08 | 96.67 |
| **AIME 2025** | | | | | | | | | | | | |
| base model | 7.13 | 11.42 | 16.68 | 22.56 | 28.89 | 35.31 | 41.64 | 48.32 | 55.61 | 62.36 | 67.27 | 70.00 |
| SFT | 22.66 | 29.02 | 34.60 | 39.70 | 44.74 | 49.52 | 53.71 | 57.32 | 60.47 | 63.10 | 64.89 | 66.67 |
| RFT | 12.15 | 14.97 | 18.43 | 23.31 | 30.36 | 39.29 | 48.31 | 56.01 | 61.63 | 64.94 | 67.45 | 70.00 |
| LUFFY | 21.35 | 26.12 | 31.11 | 36.83 | 43.98 | 49.34 | 55.98 | 60.35 | 66.56 | 69.86 | 69.99 | 70.0 |
| **Prefix-RFT** | 25.98 | 29.77 | 34.68 | 40.07 | 45.37 | 50.78 | 56.75 | 62.74 | 67.80 | 71.95 | 75.39 | 76.67 |

(45.4 vs 45.5), indicating that explicit learning from the demonstration is crucial. Conversely, updating all prefix tokens without clipping (*Update All*) leads to only marginal gains (45.7) and undesirable training dynamics (*i.e.*, rollout length explosion), confirming the necessity of our entropy-based constraint.

- **Is dynamic weighting necessary?** We replace our entropy clipping with a *Static Weight* strategy (applying a constant 0.001 weight to prefix tokens, similar to UFT). This variant yields an average score of 43.8, significantly underperforming our dynamic approach (51.8). This supports our hypothesis that weighting updates by the hybrid trajectory's advantage allows the model to selectively learn from high-value prefixes.

- **Is the gain just from clipping?** To prove that our gains are not solely due to the entropy clipping technique itself, we apply our "top-20% clipping" strategy to the on-policy rollouts of a standard RFT run (*RFT + On-policy Clip*). This achieves 43.8, performs similarly to naive RFT baseline. This confirms that the performance leap is driven by the *learning from prefix*, not the clipping trick itself.

- **Comparison with DR-PO Variant:** Finally, we evaluate a variant inspired by DR-PO, where we initialize rollouts from prefixes but do not include them in the loss calculation. The difference between the DR-PO variant and Prefix-RFT (update all) is that in this variant all $N = 8$ rollouts are sampled from the same prefix. This results in poor performance (33.8), likely due to the severe distribution shift between the guided training phase and the unguided inference phase.

*Table 8.* Comprehensive ablation results on Qwen2.5-Math-7B. Prefix-RFT outperforms all variants, justifying the synergy of prefix guidance, dynamic advantage weighting, and entropy-based clipping.

| Method Variant | MATH | Olympiad | Minerva | AIME 24 | AIME 25 | AMC | Avg. |
|---|---|---|---|---|---|---|---|
| *Prefix Gradient Strategies* | | | | | | | |
| Prefix-RFT (Freeze Prefix / All-Clip) | 84.8 | 48.3 | 37.9 | 24.1 | 16.4 | 61.2 | 45.4 |
| Prefix-RFT (Update All / No-Clip) | 83.2 | 49.8 | 41.5 | 22.5 | 18.5 | 58.5 | 45.7 |
| *Method Variants* | | | | | | | |
| Static Weight (0.001) | 83.2 | 47.6 | 39.0 | 26.0 | 18.9 | 58.4 | 43.8 |
| RFT + On-policy Clip | 85.0 | 45.0 | 39.3 | 20.2 | 12.7 | 60.6 | 43.8 |
| DR-PO Variant | 66.8 | 34.4 | 34.2 | 13.8 | 10.7 | 42.8 | 33.8 |
| **Prefix-RFT (Ours)** | **88.4** | **55.7** | **40.3** | **31.8** | **26.4** | **68.2** | **51.8** |

## A.4. Principled Design and Management of Hyperparameters

In this subsection, we discuss the principles for introducing the entropy clipping and decay mechanism and link them to the aforementioned conclusions from ablation studies. Overall, these components are grounded in two core principles: *constrained and targeted optimization on off-policy data* and *position bias & curriculum learning*. Here, we outline the rationale behind these designs and provide guidelines on tuning corresponding hyperparameters.

**Entropy-based Clipping for Targeted Learning:** The primary role of clipping is to manage the huge distribution gap between the offline expert ($\pi_{\text{off}}$) and the current policy ($\pi_\theta$). As shown in Table 4, gradients from offline tokens can be orders of magnitude larger than on-policy gradients. Without constraint, the model could quickly fit to superficial features of the demonstration (e.g., response length) rather than learning from both signals. We specifically target *high-entropy* tokens in this case because high entropy indicates model uncertainty, with the policy distribution flat and the model more likely to deviate from the expert path. This should be viewed as a conservative off-policy filter rather than a correctness oracle. For a demonstration token $a^*$ with current token distribution $p$, the imitation loss is $\ell = -\log p_{a^*}$, and the target-logit gradient magnitude is $|\partial\ell/\partial z_{a^*}| = 1 - p_{a^*}$. Low-entropy tokens are therefore in two regimes: if the model already agrees with the demonstration, this gradient is near zero; if the model confidently predicts another token, updating toward $a^*$ becomes a sharp overwrite update. High-entropy selection avoids these two extremes while leaving the actual reinforcement strength to the trajectory-level advantage.

Table 9 gives a token-level view. Almost all lowest-entropy tokens already match the demonstration and contribute little target-logit gradient, while the retained high-entropy slice still carries substantial learning signal. We also find that retaining 20% of prefix tokens already preserves most of the effective prefix-side gradient contribution (Table 10); increasing the ratio mainly admits additional lower-value off-policy tokens, which aligns with the degraded training dynamics in Sec. 6.

*Table 9.* Prefix-token statistics for entropy slices. "Top-1 = demo" denotes whether the current model's most likely token matches the demonstration token.

| Prefix-token slice | Mean entropy | Mean top-1 prob. | Top-1 = demo token |
|---|---|---|---|
| Lowest-entropy 20% | 0.0035 | 0.9997 | 99.9% |
| Middle 60% | 0.6279 | 0.8083 | 79.7% |
| Highest-entropy 20% | 3.1946 | 0.2836 | 29.8% |

*Table 10.* Normalized prefix-side gradient contribution under different retained ratios. The full-prefix update is normalized to 100%.

| Retained ratio | Normalized prefix-side contribution |
|---|---|
| 20% | 77.7% |
| 50% | 98.0% |
| 80% | 99.9% |
| 100% | 100% |

**Cosine Decay for Position Bias and Curriculum Learning:** The decay scheduler serves two purposes: mitigating position

bias and creating a natural training curriculum. Due to the autoregressive nature of LLMs, tokens at the end of a sequence are sampled less frequently. By starting with long prefixes and decaying to short ones, we ensure that the model observes expert actions in different positions. This also effectively creates a smooth transition from SFT (more guidance) to RFT (autonomous exploration), mirroring the standard two-stage post-training pipeline but within a unified loop.

These hyperparameters offer interpretable control knobs for practitioners. They can be managed by monitoring simple training metrics. For example, in our case, we monitored the rollout length: if the model begins generating excessively long or non-terminating sequences, the learning signal from long R1 CoT demonstrations is likely too strong. In this case, we can *increase* the clipping strength (reduce the ratio, e.g., 50% to 20%) or *decrease* the prefix length to reweight the training signal. We found that our default settings (20% clip, cosine decay with a 0.95 length ratio) were robust across the models we tested.

