# OpenReview forum: "Blending Supervised and Reinforcement Fine-Tuning with Prefix Sampling"
_ICML.cc/2026/Conference — ICML 2026 regular_

### Official Review · Reviewer_TiQM · 2026-02-14

**Soundness:** 3
**Presentation:** 3
**Significance:** 2
**Originality:** 2
**Overall Recommendation:** 4
**Confidence:** 3

**Summary:**

This paper presents Prefix-RFT, a novel post-training framework designed to bridge the gap between Supervised Fine-Tuning (SFT) and Reinforcement Fine-Tuning (RFT). The authors address a critical limitation in current LLM optimization: SFT’s susceptibility to distribution shift (behavioral cloning) and RFT’s inherent training instability during the early stages of exploration.The core contribution is the introduction of Hybrid Trajectories, where the model starts from an expert-provided "prefix" and completes the reasoning path through online rollout. This approach is further refined by two key mechanisms: Dynamic Advantage-based Weighting, which calibrates the learning signal for prefix tokens based on the final outcome of the self-generated path, and Entropy-based Clipping. The latter is particularly noteworthy as it selectively optimizes only the top $k\%$ highest-entropy tokens within the prefix—focusing the model’s learning on critical decision points where it is most uncertain, thereby preventing rote memorization of expert traces. To ensure a smooth transition from guided imitation to autonomous reasoning, a Cosine Decay Scheduler is employed to gradually reduce the reliance on prefixes throughout the training process.Empirical results on high-difficulty mathematical reasoning benchmarks (AIME, MATH-500) demonstrate that Prefix-RFT not only achieves superior performance over standard SFT/RFT baselines but also exhibits enhanced sample efficiency compared to concurrent hybrid methods like UFT and LUFFY. The work provides a theoretically grounded yet practical solution for stabilizing reinforcement learning in complex reasoning domains.

**Compliance With Llm Reviewing Policy:**

Affirmed.

**Final Justification:**

The authors have adequately addressed my concerns in their rebuttal, and my questions have been resolved. I appreciate the thoughtful clarifications and additional insights provided. Based on this discussion, I am inclined to increase my score.

**Key Questions For Authors:**

## **Questions**

**Q1. Entropy Clipping Rationale**: The ablation study in Section 6 effectively justifies the exclusion of low-entropy tokens by demonstrating the diminishing returns of learning already-mastered patterns. However, the rationale for focusing exclusively on high-entropy tokens warrants further clarification regarding the distinction between "lack of knowledge" and "valid diversity." High entropy often occurs at critical decision points where multiple valid reasoning paths diverge; if the model considers several correct trajectories but is forced to update toward a single specific expert prefix, there is a potential risk of over-correction that suppresses the model's own logical robustness. I would like the authors to clarify how they ensure these high-entropy updates specifically target actual logical errors rather than merely enforcing the expert's stylistic or procedural preferences, and whether any qualitative analysis exists to show that this mechanism does not inadvertently penalize valid alternative solutions.

**Q2. Impact on Solution Diversity**: How does the entropy-based clipping mechanism distinguish between "high entropy due to model uncertainty/error" and "high entropy due to competing valid reasoning paths"? If the model assigns high probability to an alternative but correct logical step, forcing it to align with the expert prefix might degrade its natural reasoning robustness. Have you conducted any qualitative analysis or used metrics (e.g., pass@k with diverse prompts) to verify whether Prefix-RFT maintains or diminishes the diversity of generated solutions compared to vanilla RFT?

**Q3. Training Efficiency**: Could you provide a comparison of "Training Wall-clock Time vs. Performance"? Does the overhead of prefix processing slow down the effective training throughput compared to standard GRPO/PPO?

**Limitations:**

yes

**Strengths And Weaknesses:**

## **Strengths**
**S1. Solid Empirical Evaluation**: The paper provides a comprehensive evaluation against strong, relevant baselines, including pure RFT (DeepSeek-R1 style), SFT, and concurrent hybrid approaches like LUFFY (Yan et al., 2025) and UFT (Liu et al., 2025). The inclusion of Pass@k (k=1 to 2048) analysis is particularly strong, convincingly demonstrating that Prefix-RFT raises the reasoning capability ceiling rather than merely improving the likelihood of known solutions.

**S2. Practical Effectiveness**: The method effectively addresses the "Cold Start" problem in RFT. The analysis in Section 5 demonstrates that Prefix-RFT successfully stabilizes training and outperforms baselines even with limited demonstration data (data efficiency). The result showing that the method helps the model solve problems that pure RFT struggles with  is a compelling practical contribution.

**S3. Theoretical Grounding**: Section 2 ("A Unified View on SFT and RFT") provides a clear theoretical framework, showing that SFT and RFT can be viewed under a unified gradient update structure. This successfully justifies the motivation for blending them via weighted gradients, making the method theoretically sound even if the novelty is incremental.

## **Weaknesses**
**W1. Limited Novelty and Similarity to Concurrent Works**: The core concept of utilizing "prefix sampling" from offline demonstrations to guide exploration is not entirely novel and is shared with concurrent works.

Similarity to UFT: As the authors acknowledge, UFT (Liu et al., 2025b) also samples prefixes from demonstrations. The primary distinction claimed—replacing UFT's static scalar weight with dynamic advantage-based weighting —appears to be an incremental engineering improvement rather than a fundamental paradigm shift.

Differentiation from LUFFY: While the authors differentiate from LUFFY (Yan et al., 2025) by using entropy clipping instead of full trace mixing, the overall framework of "hybridizing offline data with online rollouts" is heavily explored in this cycle. The contribution feels more like a specific heuristic combination (Clipping + Dynamic Weighting) rather than a novel methodological proposal.

**W2. Theoretical Justification for Entropy Clipping**: The authors choose to update high-entropy tokens in the prefix, arguing these are "critical junctures" where the model needs guidance. However, in standard active learning or PPO clipping, high entropy might also imply noise or aleatoric uncertainty. Updating strictly on high-entropy tokens from an off-policy distribution could theoretically introduce high variance. A deeper theoretical justification for why high-entropy tokens are the "correct" ones to update (vs. low-entropy confident errors) is lacking.

**W3. Domain Limitation**: The evaluation is strictly restricted to mathematical reasoning benchmarks (AIME, MATH-500, etc.). While this is the standard testbed for reasoning models, it is unclear if "Prefix-RFT" generalizes to open-ended generation or code generation, where the definition of a "valid prefix" or the utility of entropy-based clipping might differ significantly.

**W4. Computational Cost**: The paper does not explicitly analyze the added computational cost. Since the method requires processing the expert prefix for every rollout to compute the hybrid trajectory, there is an inherent inference cost during training compared to standard RFT. The trade-off between this added cost and sample efficiency is not quantified.

**Formatting Issue**: The running head on each page still displays the default template text, "Submission and Formatting Instructions for ICML 2026," instead of the actual paper title. This should be corrected for the camera-ready version.

---

> ### Author Rebuttal · Authors · 2026-03-31
>
> > W1
>
> As the reviewer points out, our paper is close to the mentioned concurrent works, where our main empirical comparison is centered. We also agree that the direction of using offline data for online RL is not unique to LLM post-training, and is more generally explored in RL research. Our contribution is instead a distinct LLM post-training formulation. To further clarify, here are detailed comparisons with UFT and LUFFY.
>
> Vs. UFT: Our reproduced UFT baseline reaches 44.2, below naive RFT (45.5), whereas Prefix-RFT reaches 51.8, suggesting that a fixed global weight on the prefix is insufficient. This also matters in practice: UFT reports significant fluctuation when the static weight is tuned across [5e-4, 1e-3, 2e-3] (Tab. 3&4, App. B.3 in their paper). By contrast, Prefix-RFT makes the amount of learning from the prefix depend on the advantage and the entropy. Both are more interpretable than the static weight and are validated to be important for final performance.
>
> Vs. LUFFY: LUFFY mixes full offline traces with on-policy rollouts, whereas Prefix-RFT uses only a prefix as guidance, leading to different off-policy update strategies. LUFFY introduces a policy reshaping function $f(\pi_\theta)=\frac{\pi_\theta}{\pi_\theta+\lambda}$ for off-policy update, and sets $\lambda=0.1$. According to LUFFY's paper (Fig. 9, App. E.4), performance drops significantly when $\lambda$ varies from 0.1 to 0.05 or 0.2. However, how to select $\lambda$ is unclear. By contrast, our `entropy_clip_ratio` has a much more direct meaning, representing how much of the off-policy prefix is retained. In our ablation, increasing the retained ratio from 20% to 50% to 80% gives a clear and interpretable trend, making the control of off-policy learning more transparent.
>
> > W2 & Q1 & Q2
>
> As the reviewer notes, Sec. 6 already shows that excluding low-entropy tokens is beneficial; the remaining question is (1) why high-entropy tokens are a better default update target than low-entropy confident errors, (2) whether this could simply reflect noise, or (3) whether it penalizes valid diversity.
>
> First, the "critical junctures" phrasing in our paper follows prior work [1], but we agree that it is too strong for our actual claim and will revise it. Our entropy clip is therefore not meant to identify "100%-correct" tokens, but to act as a conservative filter for constrained off-policy prefix learning.
>
> | Prefix-type | Freq. | Top-1 prob. | Demo-token prob. | \|dL/dz_demo\| |
> | --- | --- | ---: | ---: | ---: |
> | bottom-20%, matched | 99.90% of bottom-20% | 0.9997 | 0.9997 | 0.0003 |
> | bottom-20%, mismatched | 0.10% of bottom-20% | 0.9994 | 0.0003 | 0.9997 |
> | top-20% entropy | - | 0.2836 | 0.1496 | 0.8504 |
>
> - High vs. Low Entropy: the table shows that within the bottom-20%, almost all tokens are already matched and contribute no update, while low-entropy confident errors are extremely rare and could induce large gradients. By contrast, the high-entropy tail still carries substantial learning signal and is much less sharp and more genuinely uncertain, so the demonstration acts more like a disambiguating hint than a hard overwrite. Our claim is therefore modest: in this off-policy setting, high-entropy tokens are a safer default update target than low-entropy or random ones, not because they are always uniquely "correct," but because they provide more informative and bounded correction.
>
> - Could It Be Noise? Entropy only determines which prefix tokens are eligible for learning; the actual update is still driven by the full hybrid trajectory advantage. A prefix is reinforced only when it leads to better reward.
>
> - Penalize Valid Diversity? We view pass@2048 as the most relevant proxy for whether valid solutions remain accessible after training: Prefix-RFT remains strongest at pass@2048 on AIME24/25 (96.7/76.7), above RFT and LUFFY (90.0/70.0) and SFT (76.7/66.7). Its response length also stays much shorter than SFT after training (~2k vs ~4.5k). If Prefix-RFT mainly suppressed valid alternatives, we would expect weaker large-budget pass@k (like SFT) and more SFT-like length behavior. We observe the opposite. Though this is not a token-level proof that no valid alternative is ever penalized, it argues against a broad loss of solution robustness.
>
> Thanks reviewer for the comments and we will revise our statement accordingly.
>
> [1] Beyond the 80/20 Rule: High-Entropy Minority Tokens Drive Effective Reinforcement Learning for LLM Reasoning
>
> > W3
>
> We now add code-generation experiments, where Prefix-RFT achieves encouraging results over baseline, suggesting that the method is not restricted to math-only settings. See also Reviewer `37KX, W1`.
>
> > W4 & Q3:
>
> To clarify, our method does not add an extra rollout on top of RFT. In our setting, standard RFT uses 8 on-policy rollouts per prompt, while Prefix-RFT uses 7 standard rollouts plus 1 prefix-guided rollout. Therefore, Prefix-RFT does not increase the rollout-generation budget relative to standard RFT.

---

> > ### Author Rebuttal · Reviewer_TiQM · 2026-04-02
> >
> > Thank you for the detailed rebuttal. While the responses to W3, W4, and Q3 are satisfactory, my core concerns regarding the incremental nature of the novelty (W1) and the lack of deeper theoretical justification for entropy clipping (W2) remain only partially addressed. Therefore, I will maintain my current score.

---

> > > ### Author Response · Authors · 2026-04-04
> > >
> > > Thank you for the clarification. We understand that your remaining concerns are the novelty (W1) and the deeper theoretical justification for entropy clipping (W2).
> > >
> > > On W1, our contribution is a simple yet effective off-policy prefix-learning formulation, where trajectory-level advantage and token-level entropy clipping make prefix updates selective rather than uniformly imitative. Unlike static prefix weighting (UFT), Prefix-RFT decides both **where** to learn in the prefix and **how much** to reinforce it.
> > >
> > > On W2, our response is two-fold:
> > > > "high entropy might also imply noise or aleatoric uncertainty"
> > >
> > > **Our claim is narrower than "high-entropy tokens are correct." We use entropy as a conservative off-policy filter, not a correctness oracle.** The question is *which off-policy prefix tokens provide useful supervision without causing the sharpest policy overwrite*. Entropy only determines which prefix tokens are eligible for learning; the actual reinforcement remains advantage-driven by the full hybrid trajectory. Prefix-RFT therefore does not strictly force the model toward a single expert branch.
> > >
> > > > "updating strictly on high-entropy tokens from an off-policy ... introduce high variance."
> > > > A deeper theoretical justification for why high-entropy tokens are the "correct" ... is lacking.
> > >
> > > Owing to gradient domination, we need to constrain learning from prefix tokens, and the key question is which ones provide useful supervision without causing the sharpest policy shift. Our principle is that low-entropy tokens are in two regimes: either the model already agrees with the demonstration token, or it confidently favors another token, in which case updating toward $a*$ is a strong overwrite. For variance, excluding the latter cases removes part of a natural second-moment upper bound. This can be made precise as follows.
> > >
> > > Let $e_{a*}$ denote the one-hot vector for demonstration token $a*$; for current token distribution $p_t$, the loss is $\ell_t=-\log p_{t,a*}$ and the logit-space gradient is
> > >
> > > $$
> > > g_t=\nabla_{z_t} \ell_t = p_t-e_{a*}.
> > > $$
> > >
> > > If $u_t=\alpha_t g_t$ denotes the token-level update, where $\alpha_t$ is the clipped PPO/GRPO weight, then using $\mathrm{tr}\,\mathrm{Cov}(X)=\mathbb{E}\lVert X\rVert^2-\lVert\mathbb{E}X\rVert^2\le \mathbb{E}\lVert X\rVert^2$ and the clipped-weight bound $\alpha_t^2\le C$,
> > >
> > > $$
> > > \mathrm{tr}\,\mathrm{Cov}(u_t)\le \mathbb E[\lVert u_t\rVert_2^2] \le C\,\mathbb E[\lVert g_t\rVert_2^2].
> > > $$
> > >
> > > Consider low-entropy tokens $H(p)\le h$. Since $H(p)\ge -\log(\max_i p_i)$, some token must have probability at least $e^{-h}$. If $\arg\max_i p_i=a*$, then
> > >
> > > $$
> > > \left|\frac{\partial \ell}{\partial z_{a*}}\right|=1-p_{a*}\le 1-e^{-h},
> > > $$
> > >
> > > so dropping such tokens has little effect. If instead $\arg\max_i p_i\neq a*$, then some competing token has probability at least $e^{-h}$, hence
> > >
> > > $$
> > > \left|\frac{\partial \ell}{\partial z_{a*}}\right|=1-p_{a*}\ge e^{-h},
> > > $$
> > >
> > > so this is a confident overwrite. Since $g_t=p_t-e_{a*}$ and its $a*$-coordinate is exactly $\partial \ell / \partial z_{a*}=p_{a*}-1$, its squared norm satisfies
> > >
> > > $$
> > > \lVert g_t\rVert_2^2=\sum_i g_{t,i}^2\ge g_{t,a*}^2=\left|\frac{\partial \ell}{\partial z_{a*}}\right|^2.
> > > $$
> > >
> > > So these low-entropy disagreement cases populate the large-second-moment tail, and excluding them from prefix updates removes their contribution to the upper bound above.
> > >
> > > Thus, masking low-entropy tokens is principled: **it excludes many already-aligned tokens with much smaller second moment, while also excluding the overwrite-heavy tail.** Empirically, entropy is highly long-tailed: mean entropy is 0.4718 in the bottom-80\% slice versus 3.1946 in the retained top-20\%, so the clipped 80\% remains mainly low-entropy.
> > >
> > > This can be verified with the following table; the first two rows split tokens by whether the model's top-1 prediction agrees with the demonstration token. The last two columns report $|\partial \ell / \partial z_{a*}|=1-p_{a*}$ and $\lVert g_t\rVert_2^2$. Since $\sum_i p_{t,i}^2\le1$, we have
> > >
> > > $$
> > > \lVert g_t\rVert_2^2=1-2p_{t,a*}+\sum_i p_{t,i}^2\in[0,2].
> > > $$
> > >
> > > The disagreement row has 1.14686, already above the retained top-entropy value 0.84497.
> > >
> > > | Token group | Frequency | p_demo | abs target-grad | grad second moment |
> > > | --- | --- | ---: | ---: | ---: |
> > > | bottom-80%, top-1 agrees | 84.77% of bottom-80% (67.82% of all) | 0.90124 | 0.09876 | 0.05067 |
> > > | bottom-80%, top-1 disagrees | 15.23% of bottom-80% (12.18% of all) | 0.16397 | 0.83603 | 1.14686 |
> > > | top-entropy | 20% of all | 0.14962 | 0.85038 | 0.84497 |
> > >
> > > This supports the clipping principle: the clipped bottom-80\% contains many already-aligned tokens with much smaller second moment, while its disagreement subset has larger second moment than the retained top-entropy slice. Hence high-entropy tokens remain a safer default target than low-entropy or random tokens.
> > >
> > > We will include this clarified statement and analysis in the revised version and hope it addresses your remaining concerns.

---

### Official Review · Reviewer_iK4b · 2026-03-11

**Soundness:** 3
**Presentation:** 3
**Significance:** 2
**Originality:** 3
**Overall Recommendation:** 4
**Confidence:** 3

**Summary:**

This paper presents Prefix-RFT, a hybrid post-training method for LLMs that blends supervised fine-tuning (SFT) and reinforcement fine-tuning (RFT). The core idea is to sample a prefix from an offline demonstration, have the current policy generate a continuation, and then treat the stitched sequence as a trajectory alongside standard on-policy rollouts in RFT training. To stabilize training, the paper introduces an entropy-based clipping strategy and a cosine decay scheduler. Experiments on mathematical reasoning benchmarks show improvements over baselines.

**Compliance With Llm Reviewing Policy:**

Affirmed.

**Final Justification:**

The rebuttal addressed concerns. Increasing the score.

**Key Questions For Authors:**

1. When multiple demonstrations of varying quality exist for a single problem, how to choose the demonstration?
2. Can you provide evidence that shows Prefix-RFT incentivizes reasoning capacity instead of being a warm start?

**Limitations:**

Adding a dedicated limitations paragraph covering domain scope and reliance on external demonstrations would strengthen this paper.

**Strengths And Weaknesses:**

Strengths:
1. The paper is well-structured and easy to follow.
2. The experiments are extensive, and the ablation studies are well-designed.
3. The idea of blending supervised and reinforcement fine-tuning is a clear and meaningful generalization beyond standard SFT and RFT.

Weaknesses:
1. The performance improvements are not consistent.
2. The paper assumes one demonstration per problem. But in practice, demonstrations vary in quality, style, and length. The paper never discusses how to handle multiple candidate demonstrations for a single problem, whether certain demonstrations are more suitable for prefix sampling than others.
3. A natural concern is whether the prefix is genuinely teaching the model new reasoning patterns, or whether it is simply providing a "warm start" that nudges the model toward solutions it could have found with more sampling.

---

> ### Author Rebuttal · Authors · 2026-03-31
>
> > W1
>
> Thank you for the comment.
>
> We agree that there is some **per-benchmark variation**, and we do not intend to claim that Prefix-RFT is uniformly best on every single dataset or model. Our main point is instead about the overall pattern: on the main Qwen2.5-Math-7B setting, Prefix-RFT improves the average math score from 45.5 (RFT) to 51.8; on Qwen2.5-Math-1.5B, it reaches 41.1 compared with RFT 30.0 and remains the strongest overall method.
>
> Similar trends are also observed on Llama-3.1-8B-base and Qwen3-1.7B-base. We now observe the same overall pattern on code-generation setting, where Prefix-RFT improves over RFT on MBPP+, HumanEval+, and the Coder1 test split (please see our response to `W1 & Q1` of reviewer `37KX`). Overall, Prefix-RFT gives a robust advantage over a broad set of baselines, including recent hybrid methods.
>
> We would also emphasize that these gains come from a simple design that remains easy to integrate into a standard RFT framework. In our view, this combination of broad empirical gains and methodological simplicity is an important part of the contribution.
>
> > W2 & Q1:
>
> Thank you for this important point. Although the current submission studies the one-demonstration-per-problem setting, Prefix-RFT naturally suggests a simple default when multiple candidate demonstrations are available: we could sample different demonstrations during training rather than commit to a single one. The reason is that, in Prefix-RFT, the update on prefix tokens is not fixed, but is weighted by the advantage of the resulting hybrid trajectory (prefix+on-policy continuation). Demonstrations leading to more useful continuations therefore receive stronger learning signal, while demonstrations that do not help contribute much less. In this sense, the method is designed to be less brittle to demonstration choice than approaches that treat the entire trace as equally supervised.
>
> Meanwhile, in our submission, we provided evidence that Prefix-RFT is robust to both the quantity and the quality of demonstrations. In the data-limited analysis (App. A.3, Tab. 6), the method remains strong when the number of demonstrations is substantially reduced (from 45k to 4.5k and 0.45k), and it also shows similar performance when the demonstrations are generated by models of different strengths (DeepSeek-R1-distill-32B/7B/1.5B). While this does not fully resolve the multiple-candidate setting, it suggests that Prefix-RFT does not rely on a uniquely optimal demonstration source.
>
> More broadly, we agree that choosing among multiple demonstrations is an important and very general question. It is not specific to Prefix-RFT: the same issue arises in SFT, staged SFT+RFT pipelines, and other hybrid methods that learn from offline traces. We therefore view multi-demonstration selection as part of the broader data-quality problem and an important direction for future work; if given the opportunity to revise, we would clarify this scope more explicitly.
>
> > W3 & Q2:
>
> Thank you for this important question. We agree that entirely separating “capacity improvement” from a simpler warm-start effect is difficult. However, our current evidence argues against the warm-start explanation in three natural senses.
> 1. If the gain mainly came from a standard stage-wise warm start, then the two-stage SFT+RFT baseline should already capture most of the benefit. In our matched Qwen2.5-Math-7B setting, however, Prefix-RFT still clearly outperforms SFT+RFT (51.8 vs. 48.2).
> 2. If the prefix mainly served as a rollout-level warm start that merely makes exploration easier, then using prefixes as rollout-time guidance without learning from them should remain strong. But when we freeze prefix updates, performance drops to 45.4, essentially the same as vanilla RFT (45.5).
> 3. Our large-budget pass@k analysis is designed precisely to test whether Prefix-RFT merely increases the likelihood of solutions that the model could already discover. On the harder AIME 2025 benchmark, the base model, RFT, and LUFFY all saturate at about 70.0 pass@2048, while Prefix-RFT reaches 76.67. On AIME 2024, Prefix-RFT reaches 96.67, compared with 90.00 for RFT/LUFFY and 86.67 for the base model. We therefore view the persistent gap at pass@2048 as evidence that Prefix-RFT expands the set of correct reasoning trajectories the trained policy can eventually access, rather than only providing an easier warm start.
>
> Taken together, these results suggest that Prefix-RFT does more than provide a warm-starting recipe: it encourages the policy to acquire more reusable reasoning behavior.
>
> > Limitations
>
> Thank you for the suggestion. We will add a dedicated limitations paragraph in the revised version. We will explicitly discuss the current domain scope of the evaluation, the reliance on external demonstrations, and the fact that multi-demonstration selection remains future work.

---

> > ### Author Rebuttal · Reviewer_iK4b · 2026-04-03
> >
> > I thank the authors for their detailed response. I will keep my current score.

---

> > > ### Author Response · Authors · 2026-04-04
> > >
> > > Thank you for the acknowledgement that your concerns are fully resolved.
> > >
> > > As a supplementary clarification: the shared SFT / policy-gradient gradient structure is intended as our motivation for combining prefix supervision with PPO, rather than as the main claim by itself. Prefix-RFT is best viewed as a selective recipe for mixing off-policy prefixes into PPO, where trajectory-level advantage controls update strength and entropy clipping controls which prefix tokens are updated.
> > >
> > > Meanwhile, our additional clarification is that entropy clipping is principled rather than merely an ad-hoc engineering trick. The key argument is as follows:
> > >
> > > For the demonstration token $a*$, let $z_{a*}$ denote the corresponding logit. A low-entropy prefix token is in one of two regimes: either the model already agrees with $a*$, in which case the target-coordinate gradient magnitude is small,
> > >
> > > $$
> > > \left|\frac{\partial \ell}{\partial z_{a*}}\right| = 1 - p_{a*},
> > > $$
> > >
> > > or it confidently favors another token, in which case this quantity is large and the update is a confident overwrite. Since the full logit-gradient satisfies
> > >
> > > $$
> > > \lVert g_t \rVert_2^2 \ge \left|\frac{\partial \ell}{\partial z_{a*}}\right|^2,
> > > $$
> > >
> > > and the token-level update variance admits the upper bound
> > >
> > > $$
> > > \mathrm{tr}\mathrm{Cov}(u_t) \le \mathbb{E}\lVert u_t \rVert_2^2 \le Constant *\mathbb{E}\lVert g_t \rVert_2^2,
> > > $$
> > >
> > > So these low-entropy disagreement cases populate the large-second-moment tail, and excluding them from prefix updates removes their contribution to the upper bound above. In this sense, entropy clipping acts as a conservative filter on off-policy prefix supervision. We will revise the final version to include this theoretical clarification more explicitly.
> > >
> > > **Please refer to our Reply Rebuttal Comment for TiQM for detailed theoretical justification and relevant emperical evidence.**

---

### Official Review · Reviewer_KApg · 2026-03-13

**Soundness:** 3
**Presentation:** 3
**Significance:** 3
**Originality:** 3
**Overall Recommendation:** 4
**Confidence:** 3

**Summary:**

The paper studies the limitations of traditional post-training methods for LLMs. Supervised fine-tuning (SFT) often suffers from poor generalization since it mainly performs behavior cloning on demonstration data. Reinforcement fine-tuning (RFT), on the other hand, may learn unexpected behaviors and is highly dependent on the initial policy. To address these issues, the authors propose Prefix-RFT, which combines the two methods. The approach uses prefix sampling to leverage offline demonstrations to guide RFT-style training, where the model generates continuations and is optimized with a hybrid loss that combines imitation learning and reinforcement learning. Experimental results show improvements over standard LLM post-training methods.

**Compliance With Llm Reviewing Policy:**

Affirmed.

**Final Justification:**

The paper introduces a novel method that combines the advantages of SFT and RFT. However, the overall quality of the paper still has room for improvement. In addition, I remain concerned about the practicality of the method in real-world applications. Therefore, I would lean toward a weak accept, but I would not object if the paper is rejected.

**Key Questions For Authors:**

Please refer to the Weaknesses section above.

**Limitations:**

The paper does not discuss the limitations or potential negative impacts of the proposed method. I suggest the authors include a discussion on possible limitations, such as whether the method introduces additional computational overhead or has limited impact on improving LLM post-training.

**Strengths And Weaknesses:**

Strengths:

- The topic studied in the paper is practical, useful, and timely.

- The paper conducts comprehensive experiments across multiple baselines.

- The paper provides clear and informative visualizations of the results.

Weaknesses:

- The overall structure of the paper is somewhat messy. The paper spends too much space introducing SFT and RFT, while the related work is presented in an unexpected location.

- I am not an expert in LLM training, but I wonder how significant the impact of this hybrid method is on LLM training. Is the performance comparable to more state-of-the-art training methods?

---

> ### Author Rebuttal · Authors · 2026-03-31
>
> > W1
>
> Thank you for this comment. We agree that the paper structure could be made clearer. We would like to clarify that the discussion of SFT and RFT in Sec. 2 is not only introducing background. Rather, it serves as the core motivation for our method in two ways:
> - it places SFT, policy gradient, and PPO under a common token-weighted gradient view;
> - second, this unified view motivates using advantage to drive prefix learning, which in turn leads directly to our hybrid objective and Prefix-RFT.
> Without making these connections explicit, the method can appear more heuristic than it actually is.
>
> The placement of related work is mainly due to space constraints. In the current submission, we included a brief discussion at the end of Sec. 3 and moved the more detailed comparison with concurrent hybrid methods to App. A.1. If given the opportunity to revise, we would be happy to make the adjustments accordingly.
>
> > W2
>
> Thank you for raising this important question. We agree that it would help to better position Prefix-RFT relative to the post-training recipes commonly used in practice. Today, strong reasoning models are typically built from some combination of three paradigms: pure SFT, pure RL / RFT, or a staged pipeline that first performs SFT and then RL. This pattern is also reflected in recent technical reports on frontier reasoning models [1, 2, 3, 4]. Our point is not that Prefix-RFT is directly comparable to those systems end-to-end, since they use very different scales of data, models, and compute, but rather that Prefix-RFT improves upon these mainstream post-training paradigms in a matched setting.
>
> Concretely, in our Qwen2.5-Math-7B setting, Prefix-RFT improves the average math score from 45.5 for vanilla RFT and 48.2 for the standard two-stage SFT+RFT baseline to 51.8. It also outperforms recent hybrid baselines including UFT, ReLIFT, and LUFFY in the same setup. The same pattern appears on additional model families. We also find that Prefix-RFT remains strongest at large sampling budgets and improves pass@2048. Encouragingly, we also now observe the same overall pattern on code-generation tasks (see also our detailed response to Reviewer 37KX): Prefix-RFT consistently improves over RFT on MBPP+, HumanEval+, and the Coder1 test split, which further supports that the method is not specific to math-only settings.
>
> From a practical perspective, Prefix-RFT can also be integrated into existing RL training frameworks with minimal modifications, which makes it straightforward to adopt in practice.
>
> [1] DeepSeek-R1: Incentivizing Reasoning Capability in LLMs via Reinforcement Learning
> [2] Kimi K2.5: Visual Agentic Intelligence
> [3] GLM-5: From Vibe Coding to Agentic Engineering
> [4] Step 3.5 Flash: Open Frontier-Level Intelligence with 11B Active Parameters
>
> > Limitations
>
> Thank you for the suggestion. We agree that the paper would benefit from an explicit limitations paragraph. In the revised version, we will clarify the current scope of validation, the reliance on external demonstrations, and that Prefix-RFT does not introduce extra rollout-generation budget relative to standard RFT.

---

> > ### Author Rebuttal · Reviewer_KApg · 2026-04-03
> >
> > I thank the authors for answering my questions. Based on the responses, I will keep my evaluation as is.

---

### Official Review · Reviewer_37KX · 2026-03-15

**Soundness:** 4
**Presentation:** 3
**Significance:** 3
**Originality:** 3
**Overall Recommendation:** 4
**Confidence:** 3

**Summary:**

This paper proposes Prefix-RFT, a hybrid LLM post-training framework unifying Supervised Fine-Tuning (SFT) and Reinforcement Fine-Tuning (RFT). It samples prefixes from expert demonstrations to guide online generation, forms hybrid trajectories for standard RFT optimization, and adopts entropy-based clipping and cosine decay scheduler to stabilize training. Experiments on math and general reasoning benchmarks across multiple model scales show it outperforms standalone SFT/RFT and mainstream baselines including LUFFY and UFT

**Compliance With Llm Reviewing Policy:**

Affirmed.

**Final Justification:**

The author has resolved most of my concerns, so I’ve decided to keep my current weak accept score as is.

**Key Questions For Authors:**

1. If the authors consider applying Prefix-RFT to other downstream tasks such as code generation in the future, do they think there will be any necessary minor adjustments to the core design of the method?
2. Would the authors consider supplementing the statistics of token entropy distribution during the training process in the updated version? This may help readers more intuitively understand the selection logic of the top-k parameter, and whether this parameter needs to be adjusted adaptively for different task types.

**Limitations:**

The authors should include a discussion of limitations in their paper.

**Strengths And Weaknesses:**

Strength:
1. It establishes a unified theoretical view of SFT and RFT optimization, with an extremely simple design requiring minimal modifications to standard RFT pipelines for high engineering practicability.
2. The evaluation is comprehensive, covering diverse model scales/architectures, with systematic ablations validating core module effectiveness and result robustness.
3. In-depth analysis reveals the method’s dynamic transition between imitation and exploration, clarifying its working mechanism on problems of varying difficulty.

Weakness:
1. The current work focuses its experimental validation on reasoning tasks, and has not yet explored the performance of the method in other common LLM downstream tasks such as code generation. If there could be a brief discussion of the method’s potential in these scenarios, it would further enrich the completeness of the work.
2. Although the paper has conducted sufficient ablation experiments on the top-k entropy clipping parameter and verified the effectiveness of the default setting, it does not provide an intuitive presentation of the token entropy distribution during the training process. A supplementary statistical display of this distribution would help readers more intuitively understand the logic behind the top-k parameter selection.

---

> ### Author Rebuttal · Authors · 2026-03-31
>
> > W1 & Q1
>
> |  | mbpp+ | humaneval+ | coder1-test |
> | --- | ---: | ---: | ---: |
> | base | .359 | .378 | .108 |
> | sft | .063 | .044 | .007 |
> | sft+rl | .500 | .447 | .316 |
> | rft | .613 | .649 | .404 |
> | **prft** | **.640** | **.689** | **.429** |
>
> Thanks for the comment. To directly address it, we ran code-generation experiments. We use Qwen3-1.7B-Base, Coder1 training data with GPT-5.4-generated demonstrations, and evaluate on MBPP+, HumanEval+, and the Coder1 test split using avg@4. We umploy the same training recipe as in our main experiments. Prefix-RFT consistently improves over RFT on all three code benchmarks.
>
> Notably, SFT performs poorly in this setting, and even SFT+RL remains below RFT. Our intuition is that the GPT-5.4 demonstrations are condensed expert traces and may not be ideal full-trace imitation targets for a small 1.7B model. In other words, directly forcing the model to imitate the entire demonstration can be suboptimal, whereas Prefix-RFT only uses the demonstration as a hint with an advantage-driven and entropy-clipped update. This selective learning on demonstrations is what makes Prefix-RFT more effective.
>
> Importantly, our method does not rely on math- or reasoning-specific structure: unlike approaches such as VinePPO[1], which segment a solution into explicit reasoning steps to estimate per-step values, Prefix-RFT simply samples a prefix by randomly truncating the demonstration.
>
> > W2 & Q2
>
> We agree that how to select the hyper-parameter k could be made clearer. The key point is that k serves to control how much off-policy prefix signal is allowed into the update: if too many demonstration tokens are retained, their gradients can dominate the on-policy learning signal. This rationale is presented in the current submission. Specifically, our entropy-based clipping is applied *only* to prefix tokens, because the concern here is off-policy prefix domination (App. A.3 (Tab. 2)).
>
> We would clarify that the entropy static statistics mainly explain why clipping is sensible (see 2nd table below); the exact choice of k is actually determined by the *training dynamics and gradient analysis*:
> - Our ablations in Sec. 6 provide the practical intuition for choosing k. When we retain too many prefix tokens (e.g., top-80% and top-50%), the model quickly shifts toward the long-thinking style of the demonstrations from DeepSeek-R1, which is visible as rapid response-length growth. In practice, this response-length drift is the main signal we use to tune k: if the model starts switching too quickly into that demonstration style, k should be reduced.
> - Another useful observable is the gradient contribution from the retained prefix tokens. For example, if we normalize the effective prefix grad contribution at `k=1.0` as 100%, we find that retaining only 50% or 80% of prefix tokens already produces almost the same effective prefix-side update as keeping the full prefix:
>
> | Retained ratio `k` | Normalized prefix grad contribution |
> | --- | --- |
> | 0.2 | 77.7% |
> | 0.5 | 98.0% |
> | 0.8 | 99.9% |
> | 1.0 | 100% |
>
> The main takeaway is that (1) `k=0.2` already preserves most of the effective prefix-side update signal; (2) setting `k` as 0.5 or 0.8, the retained prefix update is already almost identical to the full-prefix case, so we directly set `k` as 0.2 for our main experiments.
>
> In the revised version, we plan to add a direct statistical view of prefix-token entropy to help the reader better understand how to choose this hyper-parameter. Our intended claim is not that there is a universally optimal k, but that k is an interpretable way to control how strongly off-policy prefixes influence training in a given domain.
> To directly answer the reviewer's question, our preliminary token-level analysis suggests that the entropy distribution of prefix tokens is highly skewed rather than uniform (e.g., p20/p50/p80/p95 are approximately 0.01/0.46/1.99/3.83 in our math setting). The high-entropy tail also coincides with the positions where the model is most uncertain and most likely to deviate from the demonstration:
>
> | Prefix-token slice | Mean entropy | Mean top-1 prob. | Top-1 = demo token |
> | --- | --- | --- | --- |
> | lowest-entropy 20% | 0.003 | 0.9997 | 99.9% |
> | middle 60% | 0.6279 | 0.808 | 79.7% |
> | highest-entropy 20% | 3.195 | 0.2836 | 29.8% |
>
> In the revised version, we will add prefix-token entropy statistics to make the choice of this hyper-parameter more transparent, although encouragingly the same setting already works well in our preliminary code-generation experiments.
>
> > Limitations
>
> Thank you for the suggestion. In the revised version, we will explicitly discuss the current domain scope of our experiments, the dependence on external demonstrations, and the fact that Prefix-RFT may still require modest task-specific tuning when applying to different domains.
>
> 1. VinePPO: Refining Credit Assignment in RL Training of LLMs

---

> > ### Author Rebuttal · Reviewer_37KX · 2026-03-31
> >
> > I thank the Authors for their response. I have no further questions, and I will keep my score unchanged.

---

> > > ### Author Response · Authors · 2026-04-02
> > >
> > > Dear Reviewer 37KX,
> > > Thank you for confirming that your concerns are fully resolved!
> > >
> > > We would be deeply grateful if you might consider raising your score to reflect this, but we respect your evaluation regardless. Thank you again!

---

### Decision · Program_Chairs · 2026-04-30

**Decision:**

Accept (regular)

**Comment:**

The submission studies LLM post-training, proposing Prefix-RFT: a hybrid method that blends Supervised Fine-Tuning (SFT) and Reinforcement Fine-Tuning (RFT) through prefix sampling from demonstrations. The authors first pointed out the problems of SFT from cloning of problematic behavior and the problems of RFT from learning of unexpected behavior, then providing a hybrid approach.

Some reviewers raise novelty concerns pointing out that Prefix-RFT is close to concurrent hybrid approaches. The main different could be seen incremental and theoretical justification is not sufficient. Some of these concerns are addressed during the rebuttal phase.

On the other hand, reviewers agree that the paper has practical value with a central idea: Prefix-RFT offers a simple way to combine guidance from demonstrations with exploration, while requiring only modest changes to a standard RFT pipeline. Reviewers also agree that experimental results are strong by showing broad comparisons, ablations, and analyses (e.g., pass@k, data efficiency, and robustness to the quality of demonstration). The author response further strengthened the claims by code-generation experiments and clarifications against standard RFT and staged SFT/RFT.

All in all, this paper is practically useful. For final version, we encourage the authors to sharpen the distinction from concurrent methods, addressing some of the major concerns and questions during the discussion phase. Those revvision could make the contribution more solid for its practical relevance.